# Somatic mutations in 3929 HPV positive cervical cells associated with infection outcome and HPV type

Maisa Pinheiro[1], Nicolas Wentzensen[1], Michael Dean [1,2], Meredith Yeager [1,2,3], Zigui Chen[4], Amulya Shastry [1,2], Joseph F. Boland[1,2], Sara Bass[1,2], Laurie Burdett[1,2], Thomas Lorey[5], Sambit Mishra [1,2], Philip E. Castle[1,6], Mark Schiffman[1], Robert D. Burk [7,8], Bin Zhu [1,9] & Lisa Mirabello[1,9] ✉

Invasive cervical cancers (ICC), caused by HPV infections, have a heterogeneous molecular landscape. We investigate the detection, timing, and HPV type specificity of somatic mutations in 3929 HPV-positive exfoliated cervical cell samples from individuals undergoing cervical screening in the U.S. using deep targeted sequencing in ICC cases, precancers, and HPV-positive controls. We discover a subset of hotspot mutations rare in controls (2.6%) but significantly more prevalent in precancers, particularly glandular precancer lesions (10.2%), and cancers (25.7%), supporting their involvement in ICC carcinogenesis. Hotspot mutations differ by HPV type, and HPV18/45-positive ICC are more likely to have multiple hotspot mutations compared to HPV16-positive ICC. The proportion of cells containing hotspot mutations is higher (i.e., higher variant allele fraction) in ICC and mutations are detectable up to 6 years prior to cancer diagnosis. Our findings demonstrate the feasibility of using exfoliated cervical cells for detection of somatic mutations as potential diagnostic biomarkers.

Invasive cervical cancer (ICC) is the fourth most common cancer worldwide[1] and virtually all cases are caused by an infection with one of the 13 high-risk (HR) human papillomavirus (HPV) types[2]. The natural history of HPV leading to ICC is well-established, mostly based on HPV16 and squamous cell carcinoma (SCC), and is characterized by a multistage disease model that starts with HPV infection, that is persistently detectable over time when not controlled by the immune system[2]. These persistent infections often lead to the development of precancerous lesions that grow within the epithelium, often for years, that eventually can invade the surrounding tissue to become ICC[2]. The

natural history of adenocarcinoma (ADC), the second most common histologic subtype, remains poorly understood.

The cancer genome atlas (TCGA) project has identified driver mutations that presumably lead to ICC[3,4]. However, ICC is a heterogeneous disease with distinct somatic mutation spectrums related to SCC and ADC histologies[4]. Recurrent somatic mutations in *PIK3CA*, *FBXW7*, *MAPK1*, *PTEN*, *EP300*, *NFE2L2*, *CASP8*, *STK11*, *HLA-A*, and *HLA-B* are enriched in SCC, while in ADC, *ELF3*, *CBFB*, *KRAS* and *ARID1A* are enriched[4,5]. One study also noted that the epigenomic and transcriptomic landscape of ICC differed by HPV species groups (*Alpha 9*

[1]Division of Cancer Epidemiology and Genetics, National Cancer Institute, National Institutes of Health, Rockville, MD, USA. [2]Cancer Genomics Research Laboratory, Leidos Biomedical Research, Inc., Frederick, MD, USA. [3]Department of Biology, Hood College, Frederick, MD, USA. [4]Department of Microbiology, The Chinese University of Hong Kong, Hong Kong, China. [5]Regional Laboratory and Women's Health Research Institute, Division of Research, Kaiser Permanente Northern California, Oakland, CA, USA. [6]Division of Cancer Prevention, National Cancer Institute, National Institutes of Health, Rockville, MD, USA. [7]Department of Epidemiology and Population Health, Albert Einstein College of Medicine, Bronx, NY, USA. [8]Department of Obstetrics & Gynecology and Women's Health, Albert Einstein College of Medicine, Bronx, NY, USA. [9]These authors contributed equally: Bin Zhu, Lisa Mirabello. ✉e-mail: mirabellol@mail.nih.gov

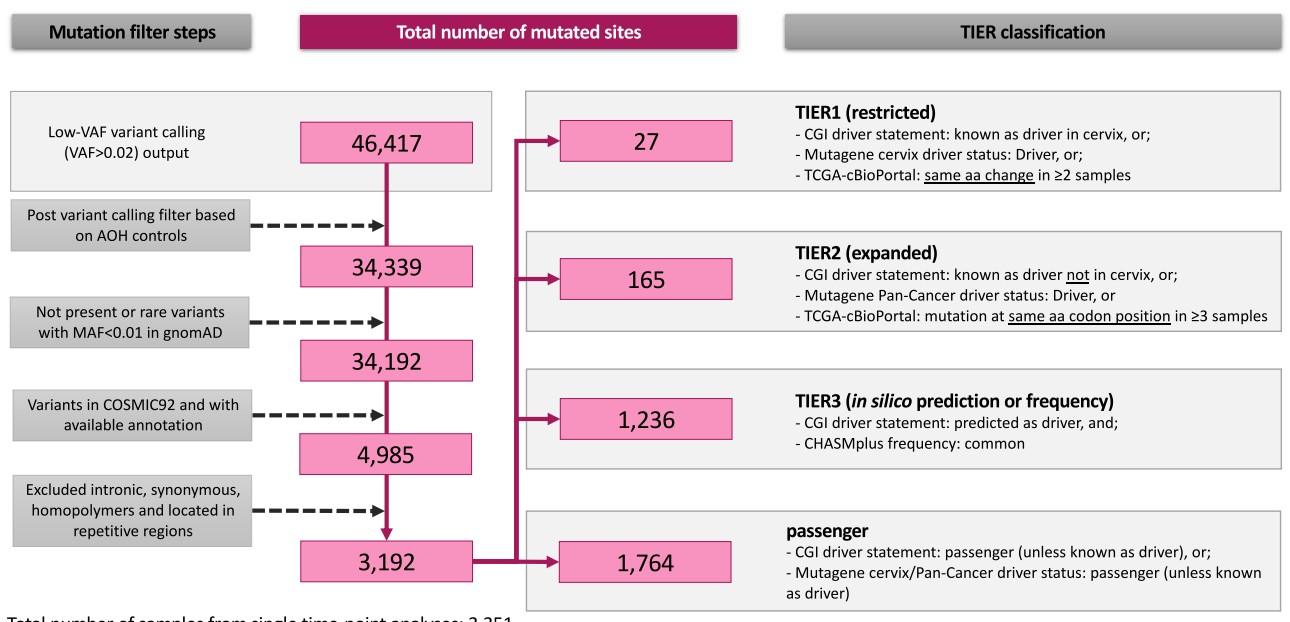

**Fig. 1 | Workflow of mutation filters and the TIER classification scheme.** Footnote: Samples from single time-point analyses only. aa = amino acid.

vs. *Alpha 7*)[6]. ICC is additionally enriched with somatic mutations induced by the off-target activity of APOBEC3 enzymes, responsible for inducing C to T or C to G changes at specific trinucleotide motifs (5′TCW3′ [W is A or T]), in response to the exogenous DNA from HPV infection[7–10]. The two most frequent mutational hotspots in ICC, E542K and E545K in *PIK3CA*, are linked to APOBEC3 activity[4,5,11]. If the intended anti-viral activity of APOBEC3 does not lead to viral clearance, it is postulated that the off-target somatic mutations may instead help drive progression to precancer and cancer[12], although, it is still unclear what triggers or promotes off-target APOBEC3 activity. The established cervical carcinogenic model, with a well-defined initiation event of HPV infection, represents a valuable opportunity to investigate when in the multi-step model somatic mutations arise and which of them drive carcinogenesis.

Different HR-HPV types, defined by ≥ 10% DNA sequence difference in the viral L1 gene Chen[13,14], are linked to profound differences in both risk and prevalence of ICC and its histological subtypes. The three most common HPV types detected in ICC worldwide are HPV16, HPV18 and HPV45[15]. HPV18 and HPV45 are genetically similar, with 74% sequence homology and both are part of the *Alpha 7* species group, while HPV16 is more genetically distant, with 52% sequence homology to both HPV18 and HPV45, and it is part of the *Alpha 9* species group[2,16]. Within each HR-HPV type there are lineages and sublineages, defined by 0.5–9% DNA sequence difference, and even finer genetic variants, that have been further linked to differences in precancer/cancer risk and lesion histology[17–19]. HPV16 is the most common cause of all ICC worldwide, including 62% of the SCC and 56% of the ADC, while HPV18 and HPV45 combined are relatively more common among ADC (43%) than among SCC (15%)[15,20,21]. Somatic mutations in *PIK3CA* are more frequent in SCC compared to ADC and in HPV16-positive tumors compared to HPV18 and HPV45 tumors[11]. It is unknown if the different mutation patterns observed across tumors are primarily linked to differences in histology or to the different associated HPV types and lineages/variants, or both.

In this study we took advantage of the Persistence and Progression (PaP) cohort, which collected residual exfoliated cervical cell samples from women routinely screened for cervical cancer precursors at Kaiser Permanente Northern California (KPNC), to investigate host somatic hotspot mutations (i.e., previously reported cancer driver mutations) by deep targeted sequencing (average coverage 820x). We utilized these residual exfoliated cervical cell samples to evaluate important driver mutations, not only in cancer samples, but also in both precancers and transient HPV infections (<cervical intraepithelial neoplasia grade 2 [CIN2] or subsequently cleared infections) at a very low variant allele fraction (VAF). We also evaluated differences in somatic mutations associated with different HPV types and APOBEC3.

## Results

### Somatic hotspot mutations are detected only in HPV-positive exfoliated cervical cells

We detected previously reported cervical cancer driver mutations among 3351 HPV-positive exfoliated cervical cell samples. A total of 3192 nucleotide *loci* were detected with one or more mutations (i.e., mutated sites) after quality control and somatic mutation filters in these single time-point samples (one sample per woman; Table S1), including 27 TIER1 mutated sites, 165 TIER2, 1236 TIER3, and 1764 passenger mutated sites (Fig. 1, Table 1, Supplementary Data 2). The VAF of mutations detected at TIER1, TIER2, TIER3 and passenger classified sites are illustrated in Fig. S2; TIER3 and passenger mutations likely included rare germline mutations with a high VAF (>0.50). Therefore, we focused on TIER1 and TIER2 mutations, and detected 176 and 784 total hotspot mutations at these TIER1 and TIER2 sites in 14 of the 20 genes sequenced (Fig. 2a, Table 1). Mutated sites in *PIK3CA*, *FBXW7* and *KRAS* were most common in TIER1, while mutated sites in *TP53* and *PTEN* were most common in TIER2 (Fig. 2a). Twenty-two percent of the TIER1 mutated sites (6 of 27 sites) and 9.1% of the TIER2 sites (15 of 165) were APOBEC3-associated mutations. In our 144 ICC samples, somatic hotspot mutations were detected at 55% of the TIER1 sites and 15% of TIER2 sites.

Only the hotspot mutations at TIER1 sites were distinctly distributed across disease status groups (Fig. 2b). Specifically, hotspot were detected in 33.3% of SCC, 21.4% of ADC, 4.5% of CIN3, 10.2% of AIS, 3.1% of CIN2, and 2.6% of controls. Of the hotspot mutations detected in the ICC samples, the most common was *PIK3CA* E545K (15.2% of all ICC), for each histology (11.4% of ADC, 20.6% of SCC), and by HPV type (16.5% of HPV16-positive ICC, 11.1% of HPV18-positive, 9.1% of HPV45-positive). Hotspot mutations *FBXW7* R505G (2.9%) and *STK11* c.290+1 G > A (2.9%) were the next most common in ADC, while for SCC, the next most common mutations were *EP300* D1399N (4.8%) and *MAPK1* E322K (4.8%) Fig. 3.

We further evaluated whether somatic hotspot mutations were present in 32 HPV-negative exfoliated cervical cell samples. No somatic hotspot mutations were detected, and only one presumed germline heterozygous TIER2 mutation was found (*TP53* R175C) with a VAF of 0.53.

### Frequency of hotspot mutations differ by HPV type

First, we evaluated the distribution of hotspot mutations among single HPV16, HPV18, and HPV45 infections only (i.e., HPV co-infected samples were excluded). Among hotspot mutations, the distribution of TIER1 mutations was significantly different between HPV16-, HPV18-, and HPV45-positive samples ($p = 0.01$; Fig. 2c). In particular, *PIK3CA* mutations were more common in HPV16-positive samples (47.2% of mutations) compared to both HPV18-positive (33.3%) and HPV45-positive (26.7%) samples; this pattern was consistent for both SCC and ADC. In

contrast, *FBXW7* mutations and *KRAS* mutations were less common in the HPV16-positive samples (*FBXW7*: HPV16 19.2% *vs.* HPV18 25.0% and HPV45 26.7% of mutations; *KRAS*: HPV16 4.8% vs. HPV18 29.2% and HPV45 6.7% of mutations, respectively). The distribution of mutations in HPV18- and HPV45-positive samples were more similar to eachother ($p = 0.21$) than either were to HPV16 ($p \leq 0.1$). The distributions of TIER2 mutations were similar among HPV types ($p = 0.54$; Fig. 2c).

The mean number of hotspot mutations per ICC sample was 1.43 (range of 1–4). HPV18/45-positive ICC were 11-fold more likely to have ≥2 hotspot mutations than HPV16-positive ICC ($p = 5.6 \times 10^{-3}$, OR = 11.2, 95%CI = 2.0–61.9), while between histologies the hotspot mutation counts were not significantly different (Table 2). HPV18/45-positive ADC were also associated with ≥2 hotspot mutations compared to HPV16-positive ADC ($p = 0.04$, OR = 7.9, 95%CI = 1.1–56.1), but not for SCC, although this is likely due to the small number of HPV18/45-positive SCC with a HS mutation ($N = 1$; Table 2).

### Hotspot mutations are progressively enriched in CIN3/AIS precancers and cancers, and influenced by viral genetic variation

We evaluated the occurrence of hotspot mutations in samples collected either at the time of or within 2 years of the case/control diagnosis ($N = 3031$; Table S1). Compared to controls, the frequency of TIER1 hotspot mutations was similar in CIN2 ($p = 0.59$), while statistically significantly increased in CIN3/AIS precancers and highest in ICC, overall and by squamous and glandular histologies: CIN3 and AIS were 2 and 4-fold more associated with TIER1 hotspot mutations, and SCC and ADC were 18 and 10-fold more associated with TIER1 hotspot mutations than controls, respectively (Table 3). The APOBEC3-associated TIER1 hotspot mutations were more strongly associated with ICC compared to controls ($p = 2.5 \times 10^{-16}$, OR = 32.5, 95% CI = 14.1–74.8) than TIER1 hotspots at non-APOBEC3 motifs ($p = 1.4 \times 10^{-7}$, OR = 6.7, 95%CI = 3.3-13.6; Table 4). Since *PIK3CA* was the most mutated gene in our cohort, we further evaluated whether it was driving the associations between ICC and hotspot mutations. Among TIER1 hotspots, *PIK3CA* mutations were 39-fold more associated with ICC ($p = 2.2 \times 10^{-16}$, OR = 38.9, 95% CI = 16.3–92.7) compared to controls, while non-*PIK3CA* mutations were 6-fold more associated with ICC ($p = 1.3 \times 10^{-6}$, OR = 5.9, 95%CI = 2.9–12.2) compared to controls (Table 3). These findings indicate that *PIK3CA* is a key driver of cervical carcinogenesis, nevertheless other mutations play a significant role.

For all TIER2 hotspot mutations, the total frequencies in controls and cases were similar (Table S5). However, both the specific TIER1 and TIER2 hotspot mutation sites that were observed in our ICC cases were rarely observed in the controls (TIER1: 1.8% of controls *vs.* 25.7% of ICC, $p < 2.2 \times 10^{-16}$; TIER2: 4.0% *vs.* 15.3% of ICC, $p = 8.2 \times 10^{-7}$) (Table S6).

**Table 1 | Distribution of mutated sites by gene and by TIER classification**

| Gene | TIER1 | | TIER2 | | TIER3 | | PASSENGER | |
|---|---|---|---|---|---|---|---|---|
| | N | % | N | % | N | % | N | % |
| *PIK3CA* | 8 | 29.6% | 15 | 9.1% | 117 | 9.5% | 126 | 7.1% |
| *FBXW7* | 6 | 22.2% | 8 | 4.8% | 117 | 9.5% | 100 | 5.7% |
| *KRAS* | 5 | 18.5% | 7 | 4.2% | 31 | 2.5% | 35 | 2.0% |
| *PTEN* | 2 | 7.4% | 23 | 13.9% | 63 | 5.1% | 69 | 3.9% |
| *ERBB3* | 2 | 7.4% | 1 | 0.6% | 82 | 6.6% | 141 | 8.0% |
| *TP53* | 1 | 3.7% | 69 | 41.8% | 123 | 10.0% | 108 | 6.1% |
| *ERBB2* | 1 | 3.7% | 5 | 3.0% | 81 | 6.6% | 121 | 6.9% |
| *EP300* | 1 | 3.7% | 3 | 1.8% | 199 | 16.1% | 211 | 12.0% |
| *MAPK1* | 1 | 3.7% | 0 | 0.0% | 13 | 1.1% | 20 | 1.1% |
| *ARID1A* | 0 | 0.0% | 11 | 6.7% | 112 | 9.1% | 344 | 19.5% |
| *STK11* | 0 | 0.0% | 11 | 6.7% | 35 | 2.8% | 47 | 2.7% |
| *NFE2L2* | 0 | 0.0% | 5 | 3.0% | 21 | 1.7% | 59 | 3.3% |
| *PIK3R1* | 0 | 0.0% | 4 | 2.4% | 64 | 5.2% | 55 | 3.1% |
| *HRAS* | 0 | 0.0% | 3 | 1.8% | 24 | 1.9% | 35 | 2.0% |
| *TGFBR2* | 0 | 0.0% | 0 | 0.0% | 76 | 6.1% | 49 | 2.8% |
| *CASP8* | 0 | 0.0% | 0 | 0.0% | 51 | 4.1% | 61 | 3.5% |
| *ELF1* | 0 | 0.0% | 0 | 0.0% | 16 | 1.3% | 35 | 2.0% |
| *HLA-B* | 0 | 0.0% | 0 | 0.0% | 10 | 0.8% | 11 | 0.6% |
| *HLA-A* | 0 | 0.0% | 0 | 0.0% | 1 | 0.1% | 5 | 0.3% |
| *MED1* | 0 | 0.0% | 0 | 0.0% | 0 | 0.0% | 132 | 7.5% |
| **Total** | 27 | 100.0% | 165 | 100.0% | 1236 | 100.0% | 1764 | 100.0% |

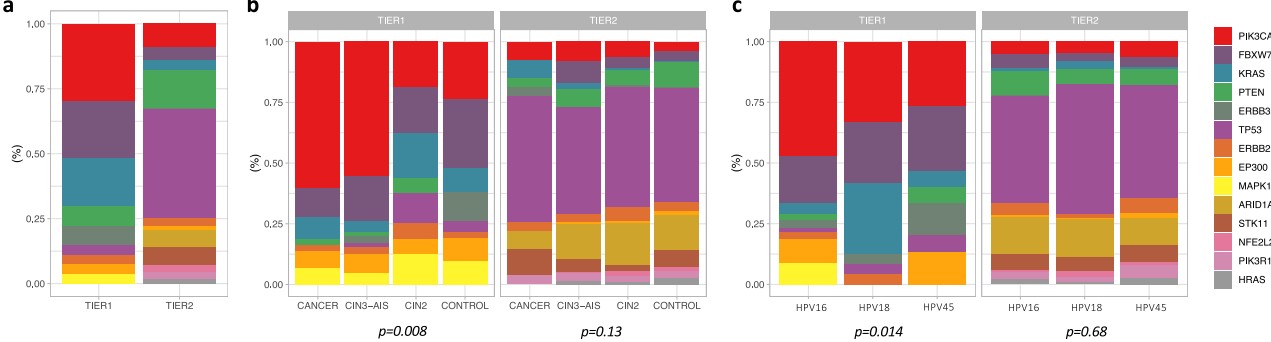

**Fig. 2 | Distribution of hotspot mutations by gene and TIER classification.** **a** Distribution of mutated sites classified in each TIER. **b** Proportion of mutations in each gene by the total number of mutations in each TIER, by status. **c** Proportion of mutations in each gene by the total number of mutations in each TIER, by HPV type. Mutations from multiple HPV16/18/45 type co-infections were excluded. Footnote:

CIN3 Cervical intraepithelial neoplasia grade 3, AIS adenocarcinoma in situ. *P*-values were estimated with two-sided Fisher's Exact tests for count data with simulated *p*-values (based on 2000 replicates). Source data are provided as a Source Data file.

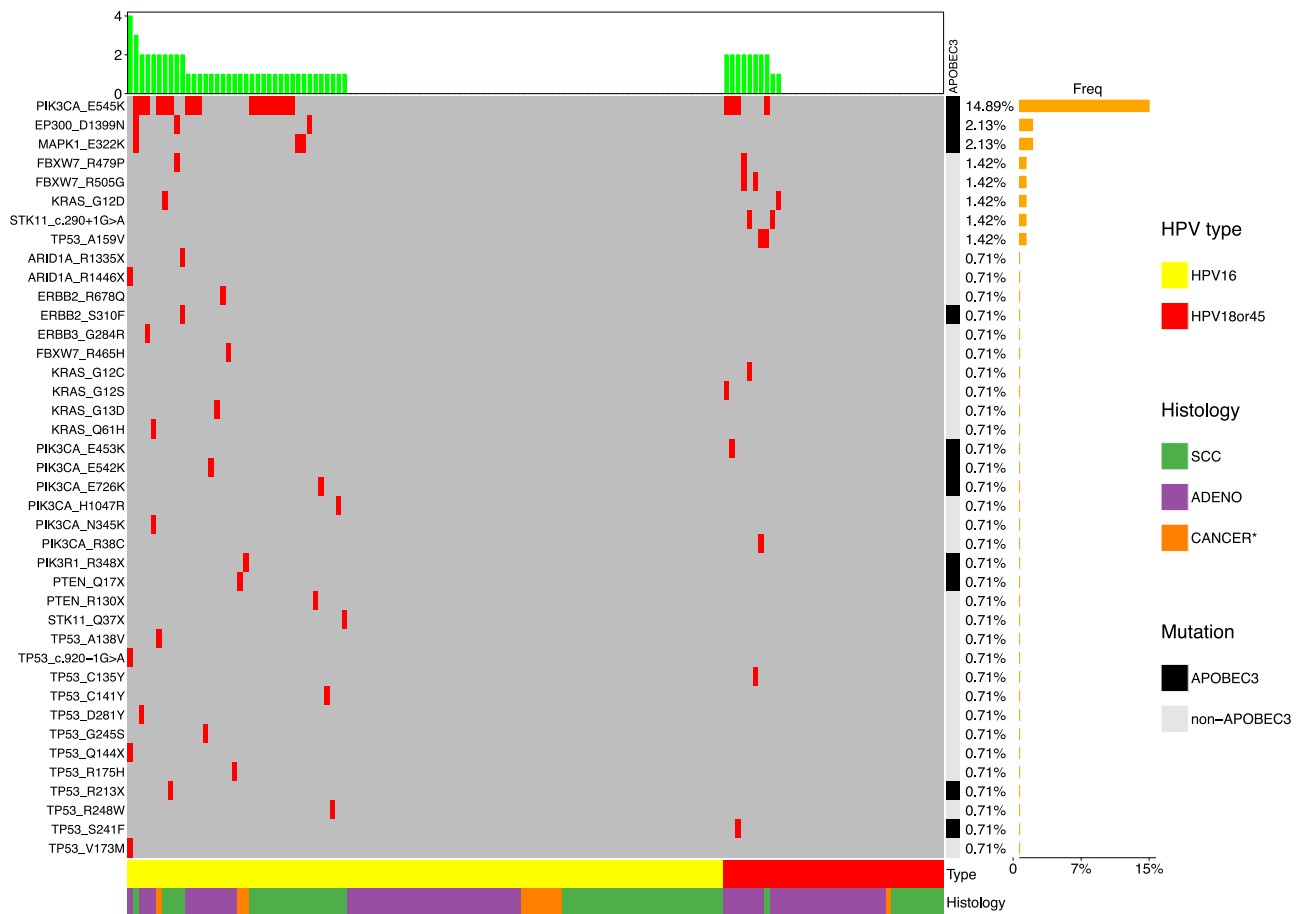

**Fig. 3 | Frequency of individual hotspot mutations in the ICC cases by HPV type and histology.** Footnote: *SCC* squamous cell carcinoma, *ADC* adenocarcinoma, Cancer* = unknown histology. Somatic mutation distribution for 141 total cancers, HPV16-positive or HPV18/45-positive, using single time point samples within 2 years of diagnosis. Source data are provided as a Source Data file.

Given the varying distribution of hotspots among HPV types/ groups, we examined the relationship between hotspot mutations and disease status with respect to HPV types/groups and within type lineages/sublineages. For HPV16-positive samples, TIER1 hotspot mutations were significantly increased in AIS precancers ($p = 8.8 \times 10^{-4}$, OR = 3.8, 95%CI = 1.7–8.5), and in both SCC ($p = 6.7 \times 10^{-16}$, OR = 19.8, 95%CI = 9.6–40.8) and ADC ($p = 8.6 \times 10^{-6}$, OR = 7.4, 95%CI = 3.0–17.9) compared to controls (Table 5). For HPV18/45-positive samples, TIER1 hotspot mutations were significantly increased in AIS ($p = 0.04$, OR = 3.4, 95%CI = 1.1–10.7) and ADC ($p = 4.3 \times 10^{-7}$, OR = 13.6, 95% CI = 4.9–37.3) compared to controls, but not significantly for squamous lesions (Table 5). Comparing cancers to controls for each HPV type and lineage/sublineage, only the previously identified 'riskier' HPV16 A4/D2/D3 sublineages[18] were more likely to be cancers with TIER1 hotspot mutations ($p < 2 \times 10^{-16}$, OR = 7.7, 95%CI = 3.5–17.3) (Fig. S3). Both APOBEC3-associated and *PIK3CA* TIER1 hotspot mutations were more enriched in ICC than the non-APOBEC3 and non-*PIK3CA* mutations, respectively, compared to controls for each HPV type/group, particularly for HPV16-positive SCC ($p = 1.5 \times 10^{-10}$; OR = 76.4, 95%CI = 20–288; Table 5, Table S7). Non-*PIK3CA* mutations were only associated with HPV16-positive SCC ($p = 2.7 \times 10^{-5}$, OR = 8.9, 95% CI = 3.1–24.8) and only with HPV18/45-positive ADC ($p = 1.7 \times 10^{-4}$, OR = 10.9, 95%CI = 3.1–37.7) (Table 5).

**Hotspot mutations were detected years before clinical diagnosis**
We leveraged the prospective aspect of this cohort to look for hotspot mutations in samples collected years before clinical diagnosis. Interestingly, we detected hotspot mutations in women whose cervical samples were collected 3 or more years before their cancer diagnosis (18% of SCC; Table S8). Compared to control samples collected within the same time-period, SCC samples collected ≥3 years prior to diagnosis had significantly more TIER1 hotspot mutations ($p = 0.02$, OR = 7.7, 95%CI = 1.3–45.2) (Table S8). The SCC cases with hotspot mutations detected in cervical cell samples collected 4 to 6 years prior to diagnosis (Table S8), had normal (WNL) or benign cytology (ASCUS) at this prior screening visit at which the hotspot was detected. For the 2 AIS and 2 ADC cases that had hotspot mutations detected 3–5 years prior to their diagnosis, one AIS had an atypical glandular cell (AGC) cytology and the other AIS and ADC cases had normal cytology at the prior screening visit where the hotspot was detected.

**Allele fractions of hotspot mutations are highest in ICC and increase over time**
We investigated whether the allele fraction of hotspot mutations differed by disease status over time prior to diagnosis. Our exfoliated cervical cell samples represent an admixed cell population, which includes both normal and tumor cells for the cases, therefore, the allele fraction could be a proxy of cellular clonal expansion. The variant allele fraction of TIER1 hotspot mutations was highest in ICC (median VAF = 0.09) compared to CIN3/AIS precancers (median VAF = 0.05, $p = 3.0 \times 10^{-4}$) and to controls (median VAF = 0.04, $p = 1.3 \times 10^{-9}$) (Fig. 4a). In addition, CIN3/AIS precancers had significantly higher TIER1 hotspot mutation allele fractions compared to controls (CIN3/AIS median VAF = 0.05 *vs.* controls median VAF = 0.04, $p = 3.4 \times 10^{-4}$). For TIER2 hotspot mutations, the allele fractions varied less by disease status, although the allele fraction in ICC (median VAF = 0.05) was

significantly higher than CIN3/AIS (median VAF = 0.04, $p = 0.02$) and controls (median VAF = 0.04, $p = 3.0 \times 10^{-4}$), but allele fractions were similar between CIN3/AIS and controls ($p = 0.09$) (Fig. S4a). These findings could indicate that some TIER2 mutations were acquired later or secondary to invasion, possibly contributing to genome instability, rather than being causal.

We also looked at this clonal expansion process from a different perspective by using multiple samples from the same women collected in a series of clinical visits. We investigated how far before cancer diagnosis a mutation would be detectable and whether the VAF of these mutations increased over time, to validate the observation presented above in different infection stages. For 396 women with an additional serial sample collected, 35 had either a TIER1 or TIER2 mutation detected in the most recent visit (TP1), including 15 ICC (34.9%), 15 CIN3/AIS precancers (6.9%), 2 CIN2 (3.2%), and 3 of the controls (4.0%) (Table S9). We then looked for these specific mutations in samples collected prior to the most recent visit (TP2, TP3, TP4 and TP5, with TP2 being the closest to TP1). We assessed the mutation VAF throughout time points for each woman. Among the 15 women with ICC and a hotspot mutation, the VAF was significantly higher in TP1 (median VAF = 0.06) than TP2 (median VAF = 0.02, paired Wilcox-test, $p = 4.9 \times 10^{-4}$) (Fig. 4b). The median VAF was also higher in TP1 for non-cancer samples, but not statistically significant (Fig. S4b). No mutations were detected in TP4 and TP5 (Fig. S4b). For this analysis, we included the TP1 mutations that were observed in the TP2-TP5 time-points at a threshold <0.02 (detailed in methods). Using a 0.02 VAF threshold for these samples, only two *PIK3CA* E545K mutations would have been detected in the TP2-TP5 time-points (Fig. S5).

**Table 2 | Hotspot mutation counts among cancers only, by HPV type/group and cancer histology**

| HPV, histology | Total | 1 HS mutation | | ≥2 HS mutations | | P | OR | 95%CI | |
|---|---|---|---|---|---|---|---|---|---|
| | | N | % | N | % | | | | |
| **HPV type/group** | | | | | | | | | |
| HPV16 | 38 | 28 | 73.7% | 10 | 26.3% | | | | |
| HPV18/45 | 10 | 2 | 20.0% | 8 | 80.0% | **5.6 × 10⁻³** | 11.20 | 2.03 | 61.89 |
| **Histology** | | | | | | | | | |
| SCC | 24 | 18 | 75.0% | 6 | 25.0% | *ref* | | | |
| ADC | 22 | 11 | 50.0% | 11 | 50.0% | 0.08 | 3.00 | 0.86 | 10.43 |
| **SCC** | | | | | | | | | |
| HPV16 | 22 | 17 | 76.2% | 5 | 23.8% | | | | |
| HPV18/45 | *1* | *0* | *0.0%* | *1* | *100.0%* | *1.00* | *-* | *-* | *-* |
| **ADC** | | | | | | | | | |
| HPV16 | 13 | 9 | 69.2% | 4 | 30.8% | | | | |
| HPV18/45 | 9 | 2 | 22.2% | 7 | 77.8% | **0.04** | 7.88 | 1.11 | 56.12 |

SCC = squamous cell carcinoma, ADC = adenocarcinoma; HS = hotspot. *P* = logistic regression; OR = odds ratio; CI = confidence interval. Significant *P* values are bolded. Results shown in italics are based on small counts and should be interpreted with caution. OR, 95%CI, and *P* values were estimated using logistic regression; tests were two-sided.

**Table 3 | Association of TIER1 hotspot mutations with precancers and cancers among single time-point samples collected within 2 years of outcome ascertainment**

| Status | Total | no HS mutation | | HS mutations | | P | OR | 95%CI | |
|---|---|---|---|---|---|---|---|---|---|
| | | N | % | N, ≥1 HS | % | | | | |
| **Control** | 1300 | 1266 | 97.4% | 34 | 2.6% | *ref* | | | |
| **CIN2** | 520 | 504 | 96.9% | 16 | 3.1% | 0.59 | 1.18 | 0.65 | 2.16 |
| **CIN3/AIS** | 1067 | 1010 | 94.7% | 57 | 5.3% | **7.7 × 10⁻⁴** | 2.10 | 1.36 | 3.24 |
| CIN3 | 909 | 868 | 95.5% | 41 | 4.5% | **0.02** | 1.76 | 1.11 | 2.79 |
| AIS | 157 | 141 | 89.8% | 16 | 10.2% | **5.0 × 10⁻⁶** | 4.23 | 2.28 | 7.85 |
| **Cancer** | 144 | 107 | 74.3% | 37 | 25.7% | **<2.0 × 10⁻¹⁶** | 12.88 | 7.76 | 21.35 |
| SCC | 63 | 42 | 66.7% | 21 | 33.3% | **<2.0 × 10⁻¹⁶** | 18.64 | 9.98 | 34.82 |
| ADC | 70 | 55 | 78.6% | 15 | 21.4% | **8.1 × 10⁻¹²** | 10.16 | 5.23 | 19.76 |
| | | N | % | N, *PIK3CA* | % | | | | |
| **Control** | 1273 | 1266 | 99.5% | 7 | 0.5% | *ref* | | | |
| **CIN2** | 507 | 504 | 99.4% | 3 | 0.6% | 0.92 | 1.08 | 0.28 | 4.18 |
| **CIN3/AIS** | 1041 | 1010 | 97.0% | 31 | 3.0% | **4.6 × 10⁻⁵** | 5.55 | 2.43 | 12.66 |
| CIN3 | 891 | 868 | 97.4% | 23 | 2.6% | **3.1 × 10⁻⁴** | 4.79 | 2.05 | 11.22 |
| AIS | 149 | 141 | 94.6% | 8 | 5.4% | **9.3 × 10⁻⁶** | 10.26 | 3.67 | 28.72 |
| **Cancer** | 130 | 107 | 82.3% | 23 | 17.7% | **2.2 × 10⁻¹⁶** | 38.88 | 16.31 | 92.68 |
| SCC | 55 | 42 | 76.4% | 13 | 23.6% | **4.4 × 10⁻¹⁶** | 55.96 | 21.24 | 147.45 |
| ADC | 64 | 55 | 85.9% | 9 | 14.1% | **8.9 × 10⁻¹¹** | 29.59 | 10.63 | 82.39 |
| | | N | % | N, non-*PIK3CA* | % | | | | |
| **Control** | 1290 | 1266 | 98.1% | 24 | 1.9% | *ref* | | | |
| **CIN2** | 517 | 504 | 97.5% | 13 | 2.5% | 0.38 | 1.36 | 0.69 | 2.69 |
| **CIN3/AIS** | 1032 | 1010 | 97.9% | 22 | 2.1% | 0.64 | 1.15 | 0.64 | 2.06 |
| CIN3 | 883 | 868 | 98.3% | 15 | 1.7% | 0.78 | 0.91 | 0.48 | 1.75 |
| AIS | 148 | 141 | 95.3% | 7 | 4.7% | **0.03** | 2.62 | 1.11 | 6.19 |
| **Cancer** | 119 | 107 | 89.9% | 12 | 10.1% | **1.3 × 10⁻⁶** | 5.92 | 2.88 | 12.16 |
| SCC | 48 | 42 | 87.5% | 6 | 12.5% | **2.9 × 10⁻⁵** | 7.53 | 2.93 | 19.40 |
| ADC | 61 | 55 | 90.2% | 6 | 9.8% | **2.4 × 10⁻⁴** | 5.75 | 2.26 | 14.65 |

CIN2 = cervical intraepithelial neoplasia grade 2; CIN3 = CIN grade 3, AIS = adenocarcinoma in situ; SCC = squamous cell carcinoma, ADC = adenocarcinoma; HS = hotspot; P = *P* value by multinomial logistic regression; OR = odds ratio; CI = confidence interval. Significant P values are bolded. OR, 95%CI, and P values were estimated using multinomial logistic regression; tests were two-sided.

**Table 4 | Associations of TIER1 hotspot mutations matching APOBEC3 and non-APOBEC3 motifs**

| HPV | Status | Total | No HS mutation | | ≥1 HS mutation | | P | OR | 95%CI | |
|---|---|---|---|---|---|---|---|---|---|---|
| All | | | N | % | N, APOBEC3 | % | | | | |
| | Control | 1274 | 1266 | 99.4% | 8 | 0.6% | ref | | | |
| | Cancer | 129 | 107 | 82.9% | 22 | 17.1% | $2.5 \times 10^{-16}$ | 32.5 | 14.1 | 74.8 |
| | | | N | % | N, non-APOBEC3 | % | | | | |
| | Control | 1289 | 1266 | 98.2% | 23 | 1.8% | ref | | | |
| | Cancer | 120 | 107 | 89.2% | 13 | 10.8% | $1.4 \times 10^{-7}$ | 6.7 | 3.3 | 13.6 |

Samples with co-occurrence of both APOBEC3 and non-APOBEC3 induced mutations were excluded from the analyses. HS hotspot, *P* = logistic regression; *OR* odds ratio; *CI* confidence interval. Significant *P*-values are bolded. OR, 95%CI, and *P*-values were estimated using logistic regression; tests were two-sided.

Lastly, we investigated whether specific mutation's VAF increased over time faster than others, suggesting that these mutations could be the leading drivers being selected. Because samples were collected in a clinical setting, the time interval between TPs varied, with a mean time interval of 1.24 years (range 0.50–5.37 years), therefore, we calculated a rate of VAF change considering the time interval. Mutations with the fastest average rates of change were *PIK3CA* E542K ($r = 0.132$ VAF increase per year) and E545K ($r = 0.129$), *ERBB2* R678Q ($r = 0.062$), *TP53* R213X ($r = 0.045$), *ARID1A* R1446X ($r = 0.034$), and *PTEN* Q17X ($r = 0.022$) (Fig. S5). The first three mutations with the fastest VAF change were all missense mutations in oncogenes, and the next three were nonsense mutations in tumor suppressor genes.

## Discussion

We have shown that hotspot mutations can be detected in exfoliated cervical cell samples collected prior to precancer/cancer clinical diagnosis at routine cervical cancer screening visits and that HPV types/groups, and within type lineages/sublineages, influence somatic mutation frequencies. Using exfoliated cervical cells and deep targeted sequencing, we were able to detect important hotspot driver mutations. These mutations were significantly more prevalent in precancers and cancers, with up to a 76-fold increase in cancers depending on mutation type and HPV type, compared with controls. *PIK3CA* and APOBEC3-induced mutations were the most common mutations detected in this cohort, and some non-*PIK3CA* mutations were also significantly associated with ICC compared with controls. We observed an increase in the allele fraction of hotspot mutations from controls (i.e., HPV transient infections: < CIN2 or subsequently cleared infections) through precancers and cancers, in line with the predicted cellular clonal expansion in cancer development.

We have identified important TIER1 mutation differences by viral genetic variation, and demonstrate that HPV type/group influences the somatic landscape. The overall distribution of TIER1 mutations and the number of mutations were significantly different between HPV16-, HPV18-, and HPV45-positive cases. Hotspot mutations in *PIK3CA* were more common in HPV16-positive cases, while mutations in *KRAS* and *FBXW7* were more common in HPV18-positive and HPV45-positive samples, respectively. The non-*PIK3CA* mutations were HPV type and tumor histology dependent. Interestingly, we also observed that the HPV16 A4/D2/D3 sublineages, which have been previously associated with an increased risk of ICC and particularly strong increased risks of ADC[18], were specifically associated with cancers having a hotspot mutation, compared to the other HPV16/18/45 sublineages. In addition, the HPV18/45-positive cancers had a higher number of mutations compared to the HPV16-positive cancers, independent of histology. The HPV18-positive ICCs were 11 times more likely to have 2 or more hotspot mutations compared to HPV16-positive ICC. It is possible that HPV16, as a potentially stronger carcinogen, may require fewer additional somatic mutations in host cells to drive carcinogenesis, and in contrast, HPV18/45 may require more mutations, although we did not evaluate other somatic events such as copy number alterations and viral integration. HPV18 and HPV45 have a higher prevalence of HPV integration than HPV16 and are likely associated with more

chromosomal damage[4,22], which further supports HPV16 being a stronger carcinogen with less associated somatic events. It's possible that the less prevalent/carcinogenic HR-HPV types could have even more somatic mutations driving carcinogenesis. Follow-up studies to characterize and compare somatic mutations in case samples positive for the less carcinogenic HR-HPV types, including HPV31 and HPV35 (the HPV16-related types), are needed.

Our comprehensive mutation classification scheme into TIER1 and TIER2, based on previously published somatic and functional data, was critical for distinguishing hotspot mutations more likely to be drivers for ICC. Only TIER1 mutations, defined as hotspots previously reported as cervical cancer drivers, were significantly associated with precancers and cancers in our cohort, demonstrating that mutations classified as somatic drivers for non-cervical cancers (TIER 2) were not as important or the main drivers of cervical carcinogenesis. TIER1 hotspot mutations were enriched in both HPV16- and HPV18/45-positive glandular precancers and cancers (AIS and ADC) compared to controls, while only HPV16-positive SCC had significantly more mutations than controls. Although, there were only 10 HPV18/45-positive SCC in our cohort and this likely limited statistical power. Differences in the somatic landscape of ICC by squamous and glandular histologic subtypes have been previously observed and related to expression profiles[4,5], however it is not clear if specific driver mutations trigger tumor differentiation differently or if the somatic landscape is influenced by HPV type. The majority of significantly mutated genes previously reported in ADC were also observed in SCC; however very few ADC samples (only 4–31 [TCGA]) were previously investigated in earlier studies[4,6,11]. Future studies focusing on larger numbers of glandular lesions may help to identify new significantly mutated genes in this subtype, and lead to a better understanding of the true differences in ICC etiology related to specific histologic subtypes and HPV types.

Importantly, our HPV-negative cervical cell samples had no TIER1 mutations, suggesting these mutations are uncommon in HPV-negative cells. Recent studies have shown that some cancer associated recurrent mutations are also identified in normal cells from the same epithelium[23,24], proposing that tissue-specific transformation likely requires additional factors such as environmental exposures, additional mutations or ineffective immune surveillance[25]. In our cohort, it is possible that some of the somatic mutations found in the HPV-positive control samples are a consequence of errors in intrinsic processes of the infected dividing epithelium, such as aberrant DNA replication or repair, and potentially related to the HPV infection; and these mutations are likely not enough to drive transformation alone.

We detect somatic driver mutations in exfoliated cervical cell specimens, which is important because these samples represent a less invasive sampling procedure using residual samples from current routine cervical cancer screening, as compared to tumor blocks or tissue biopsies. We demonstrated that these samples can be used to evaluate potential diagnostic and predictive somatic mutations. Our deep gene-panel sequencing assay with 800x mean depth allowed us to identify somatic hotspot driver mutations in these cervical cells at a

**Table 5 | Association of TIER1 hotspot mutations with precancers and cancers by HPV type/groups among samples collected within 2 years of outcome ascertainment**

| HPV type/group | Status | Total | No HS mutation | | HS mutations | | P | OR | 95%CI | |
|---|---|---|---|---|---|---|---|---|---|---|
| | | | N | % | N, ≥1 HS | % | | | | |
| HPV16 | Control | 665 | 645 | 97.0% | 20 | 3.0% | *ref* | | | |
| | CIN2 | 244 | 238 | 97.5% | 6 | 2.5% | 0.66 | 0.81 | 0.32 | 2.05 |
| | CIN3/AIS | 844 | 798 | 94.5% | 46 | 5.5% | **0.02** | 1.86 | 1.09 | 3.17 |
| | CIN3 | 750 | 714 | 95.2% | 36 | 4.8% | 0.09 | 1.63 | 0.93 | 2.84 |
| | AIS | 94 | 84 | 89.4% | 10 | 10.6% | **$8.8 \times 10^{-4}$** | 3.84 | 1.74 | 8.48 |
| | Cancer | 103 | 75 | 72.8% | 28 | 27.2% | **$4.2 \times 10^{-15}$** | 12.04 | 6.47 | 22.42 |
| | SCC | 50 | 31 | 62.0% | 19 | 38.0% | **$6.7 \times 10^{-16}$** | 19.77 | 9.58 | 40.77 |
| | ADC | 43 | 35 | 81.4% | 8 | 18.6% | **$1.0 \times 10^{-5}$** | 7.37 | 3.03 | 17.91 |
| | | | N | % | N,PIK3CA | % | | | | |
| | Control | 648 | 645 | 99.5% | 3 | 0.5% | *ref* | | | |
| | Cancer | 93 | 75 | 80.6% | 18 | 19.4% | **$5.4 \times 10^{-10}$** | 51.60 | 14.85 | 179.28 |
| | SCC | 42 | 31 | 73.8% | 11 | 26.2% | **$1.5 \times 10^{-10}$** | 76.36 | 20.25 | 287.85 |
| | ADC | 41 | 35 | 85.4% | 6 | 14.6% | **$7.3 \times 10^{-7}$** | 36.90 | 8.85 | 153.81 |
| | | | N | % | N, non-PIK3CA | % | | | | |
| | Control | 659 | 645 | 97.9% | 14 | 2.1% | *ref* | | | |
| | Cancer | 83 | 75 | 90.4% | 8 | 9.6% | **$5.3 \times 10^{-4}$** | 4.91 | 2.00 | 12.10 |
| | SCC | 37 | 31 | 83.8% | 6 | 16.2% | **$2.7 \times 10^{-5}$** | 8.92 | 3.21 | 24.78 |
| | ADC | 37 | 35 | 94.6% | 2 | 5.4% | 0.21 | 2.63 | 0.58 | 12.04 |
| HPV18/45 | | | N | % | N, ≥1 HS | % | | | | |
| | Control | 557 | 543 | 97.5% | 14 | 2.5% | *Ref* | | | |
| | CIN2 | 210 | 204 | 97.1% | 6 | 2.9% | 0.79 | 1.14 | 0.43 | 3.01 |
| | CIN3/AIS | 159 | 154 | 96.9% | 5 | 3.1% | 0.66 | 1.26 | 0.45 | 3.55 |
| | CIN3 | 108 | 107 | 99.1% | 1 | 0.9% | 0.33 | 0.36 | 0.05 | 2.79 |
| | AIS | 50 | 46 | 92.0% | 4 | 8.0% | **0.04** | 3.37 | 1.07 | 10.67 |
| | Cancer | 38 | 30 | 78.9% | 8 | 21.1% | **$1.2 \times 10^{-6}$** | 10.34 | 4.03 | 26.56 |
| | SCC | 10 | 9 | 90.0% | 1 | 10.0% | 0.18 | 4.31 | 0.51 | 36.37 |
| | ADC | 27 | 20 | 74.1% | 7 | 25.9% | **$4.3 \times 10^{-7}$** | 13.57 | 4.94 | 37.31 |
| | | | N | % | N,PIK3CA | % | | | | |
| | Control | 547 | 543 | 99.3% | 4 | 0.7% | *ref* | | | |
| | Cancer | 34 | 30 | 88.2% | 4 | 11.8% | **$7.4 \times 10^{-5}$** | 18.14 | 4.32 | 76.07 |
| | SCC | 10 | 9 | 90.0% | 1 | 10.0% | **0.02** | 15.12 | 1.54 | 148.95 |
| | ADC | 23 | 20 | 87.0% | 3 | 13.0% | **$1.5 \times 10^{-4}$** | 20.39 | 4.28 | 97.24 |
| | | | N | % | N, non-PIK3CA | % | | | | |
| | Control | 553 | 543 | 98.2% | 10 | 1.8% | *ref* | | | |
| | Cancer | 34 | 30 | 88.2% | 4 | 11.8% | **$1.4 \times 10^{-3}$** | 7.24 | 2.15 | 24.44 |
| | SCC | 9 | 9 | 100.0% | 0 | 0.0% | 0.93 | - | - | - |
| | ADC | 24 | 20 | 83.3% | 4 | 16.7% | **$1.7 \times 10^{-4}$** | 10.87 | 3.14 | 37.66 |

*CIN2* cervical intraepithelial neoplasia grade 2, *CIN3* CIN grade 3, *AIS* adenocarcinoma in situ, *SCC* squamous cell carcinoma, *ADC* adenocarcinoma. Samples with co-occurrence of both *PIK3CA* and non-*PIK3CA* mutations were excluded from the analyses. *HS* hotspot; *P* = multinomial logistic regression, *OR* odds ratio, *CI* confidence interval. Significant *P*-values are bolded. OR, 95%CI, and *P*-values were estimated using multinomial logistic regression; tests were two-sided.

low allele fraction. In our single time-point analyses, compared with controls, the allele fraction of TIER1 hotspot mutations were higher in precancers and highest in ICC. In the multiple time-point analyses, the allele fraction of hotspot mutations were also significantly higher closest to diagnosis, supporting the predicted increasing trend of the allele fraction with cellular clonal expansion. A similar study using residual samples from liquid-based cytology specimens from the thyroid, lymph node, breast, pancreas and other fluids used targeted NGS with a mean depth of 500x, and a VAF threshold of 10%, similarly identified important somatic mutations primarily in the samples with a higher proportion of tumor cells[26]. Our exfoliated cervical samples represent an admixture of normal and tumor cells. As for the allele fraction of TIER2 hotspot mutations, they were similar between controls and precancers in the single time-point analyses, but significantly higher in ICC, which may indicate that some of these mutations are

capable of driving tumorigenesis in later clones or they are less important/passengers carried along with pre-malignant cells and may contribute to genomic instability.

In our prospective serial sample collection, we also observed hotspot mutations years prior to the case diagnosis and differences in the allele fraction by time. The majority of mutations were detected within 2 years of the time of diagnosis, and, although rare, we also detected three TIER1 hotspot mutations and six TIER2 hotspot mutations between 3 and 6 years before the cancer diagnosis. These findings support that driver mutations that confer a proliferative advantage for tumors can be detected years before clinical diagnosis[27]. We may have missed more of these early mutations because residual cervical cell specimens collected many years prior to diagnosis may not be enriched enough for cancer precursor cells compared to the normal cells. The ratio of VAF changes per time-point was highest for

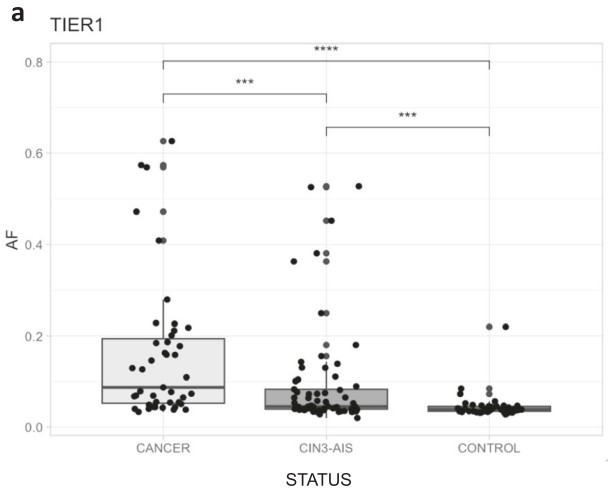

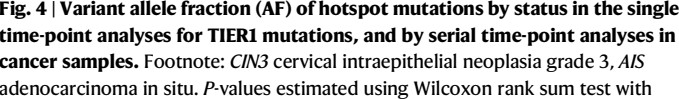

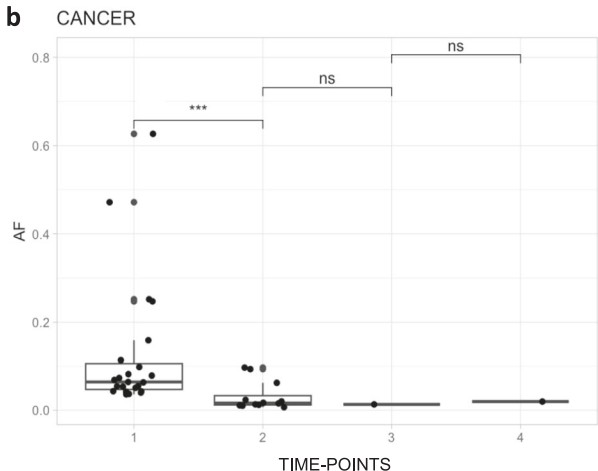

**Fig. 4 | Variant allele fraction (AF) of hotspot mutations by status in the single time-point analyses for TIER1 mutations, and by serial time-point analyses in cancer samples.** Footnote: *CIN3* cervical intraepithelial neoplasia grade 3, *AIS* adenocarcinoma in situ. *P*-values estimated using Wilcoxon rank sum test with continuity correction. Tests were two sided. ns not significant; ***p*-value ≤ 0.001; ****p*-value ≤ 0.0001. Source data are provided as a Source Data file, and provide the exact *p*-values.

*PIK3CA* E542K and E545K mutations. It is possible that clonal expansion was faster in cells harboring these mutations or that these mutations were mapped to genomic regions that were later amplified. In both scenarios, these mutations could have been selected and contributed to driving carcinogenesis. Selection of driver mutations is still not completely understood, as many coding mutations are tolerated during carcinogenesis[24,28], and more studies are needed to understand when driver mutations arise.

*PIK3CA* was the most frequently mutated gene in our cohort, and we detected strong differences in *PIK3CA* mutation frequencies between cases and controls, depending on HPV type and histology (with up to 76-fold differences in risk associations). *PIK3CA* mutations were present in 18% of our ICC samples, which is slightly lower than previously published data from tissue samples, which have been reported in 22% of tumors from Latin-American and up to 45% in HIV-negative tumors from Uganda[4,6,11]. A combination of other non-*PIK3CA* mutations including *FBXW7*, *KRAS*, *PTEN*, *ERBB3*, *TP53*, *ERBB2*, *EP3OO* and *MAPK1* were also associated with precancer and ICC, but only when classified as TIER1. APOBEC3-induced mutations are a recognized source of somatic mutations[29], and here, we confirmed TIER1 hotspot mutations consistent with being induced by APOBEC3 are enriched in our cancers but not in the HPV-positive controls. The most common *PIK3CA* hotspot, E545K, matching an APOBEC3 motif, was ~18-fold more frequent in ICC than controls. Interestingly, even though APOBEC3, as part of the host's intracellular defense, is activated upon HPV infection, its induced somatic mutations were significantly lower in both control (i.e., HPV transient infection: <CIN2 or subsequently cleared infection) and persistent HPV+ infections in our study, potentially indicating APOBEC3 mutated host DNA more in the cases. We previously observed that APOBEC3-induced viral mutations in the HPV16 genome were significantly associated with a benign infection or viral clearance[30]; together these data support a double-edged sword hypothesis, when APOBEC3 mutations in the viral genome do not lead to viral clearance, its off-target activity can instead result in host somatic driver mutations. Among HPV-positive oropharyngeal cancers, we have further shown that APOBEC3 mutations in paired HPV16 genomes and host somatic genomes were correlated, suggesting a common mechanism of APOBEC3 mutagenesis in the host and viral genomes of these tumors[31]. However, more work is needed to fully understand the combination of mechanisms that likely lead to APOBEC dysregulation and off-target mutagenesis as they are also prevalent in non-virally mediated tumors.

Our study has some limitations that should be noted. Given that our samples are from a clinical setting, DNA from matched "normal" samples was not available, therefore, we used publicly available polymorphism databases to filter out germline variants. Unfortunately, this approach likely missed rare germline variants with a lower-than-expected VAF (e.g., <50%), which could have been considered somatic. For example, the mutation in *TP53* (R175C) detected in the HPV-negative sample with an allele fraction of 0.53 is likely a rare germline variant. However, we did not want to restrict mutations to only those with a low allele fraction (e.g., <30%) because this would cause us to miss important driver mutations potentially located in amplified regions (for example at 3q[4]). Our study was not designed for discovering novel somatic driver mutations, and instead we limited our analysis to a fixed number of important genes to allow for deep targeted sequencing and only to previously reported somatic mutations. Follow-up studies including matched germline samples and evaluations to confirm the cervical tumor origin of the cervical cells are needed. Our prospective cohort includes cervical cell samples collected from women undergoing routine cervical cancer screening, therefore, the precise timing of disease diagnosis may be limited by the timing of the screening visit intervals. To account for potential undetected disease due to the screening visit intervals, we grouped the samples collected within 2 years of the date of clinical diagnosis as 'at diagnosis' samples. However, we cannot exclude that a longer time interval underlies prevalent disease.

In summary, our study has identified somatic driver mutations for cervical cancer in residual cervical cell samples that are routinely collected from a clinical setting prior to precancer/cancer diagnosis and demonstrates the feasibility of using them for detecting driver mutations that are potentially diagnostic biomarkers. We further demonstrate that HPV type and genetic variation influence the host somatic landscape and that specific somatic driver mutations are enriched in precancers and cancers compared to HPV-positive control samples (<CIN2 or subsequently cleared infections). Our deep targeted sequencing approach using cervical cells requires validation as it has the potential to be translated into a diagnostic for cervical precancer/cancer in a clinical laboratory, and could be higher throughput than full molecular profiling of cancers for treatment. Our findings demonstrate the potential of using these convenient samples to detect important somatic driver events before cancer diagnosis.

## Methods

### Study population and sample collection

The Kaiser Permanente Northern California (KPNC) institutional review board (IRB) approved use of the data, and the National Institutes of Health Office of Human Subjects Research deemed this study exempt from IRB review.

Residual exfoliated cervical cell samples were selected from women in KPNC-NCI HPV Persistence and Progression (PaP) cohort. A full and detailed description of the cohort was previously reported[32]. Briefly, the PaP cohort includes ~55,000 women out of ~1 million who underwent routine cervical cancer screening using the Hybrid Capture 2 assay (HC2; Qiagen Inc., Gaithersburg, MD) and cytology between December 2006 and January 2011. Participants could opt-out of retaining residual cervical specimens from pap-smears and those samples were discarded (~8% of women opted out). The retained residual cervical cells were stored in liquid-based specimen transport medium (STM; Qiagen Inc., Gaithersburg, MD). Women were followed over time and we obtained coded information on subsequent cervical cancer screening test results and histology results from electronic health records through 2019. All personally identifying information was kept strictly at KPNC. Histology was determined based on the cervical intraepithelial neoplasia (CIN) classification system.

For our study, we included cervical cell samples from precancer/cancer cases and controls (described below) that were positive for HPV16, HPV18 and/or HPV45 using Linear Array (LA; Roche Molecular Systems, Pleasanton, CA, USA), Cobas (Roche) and/or lab-specific polymerase chain reaction (PCR). A total of 3,351 women were included: 1,478 controls, 561 CIN grade 2 (CIN2; equivocal squamous precancer), 984 CIN grade 3 (CIN3; squamous precancer), 166 adenocarcinoma in situ (AIS; glandular precancer), 1 precancer with unknown histology, 74 SCC, 76 ADC, and 11 ICC with unknown histology (Table S1). We selected all available HPV16-, HPV18- and HPV45-positive precancer (CIN3, AIS) and cancer (ICC, SCC, ADC) samples. Eight adenosquamous carcinoma cases were included with ADC for histology analyses. Controls were defined as women having baseline HPV16-, HPV18- and/or HPV45-positive specimens that subsequently cleared their infection and/or had an infection defined as normal or low-grade lesion (ASCUS, LSIL, CIN1) with no histologic evidence of equivocal precancer or worse (CIN2+) during the study follow-up period according to data obtained from the electronic health records. 77.4% of our controls subsequently cleared their infections during the study follow-up time. At least 1 control per CIN3/AIS+ case was randomly selected for comparisons (Table S1). A subset of the total CIN2 cases available were randomly selected for inclusion. The average age of the women included in our study was 38 years (SD 11), and the majority self-reported their race/ethnicity as White (51%), followed by Hispanic (20%), Asian/PI (15%), Black (8%), or multiracial/other (7%). There were 136 women that had HPV18 and HPV16 co-infections, 21 had HPV18 and HPV45 co-infections, 84 had HPV45 and HPV16, and 9 had HPV16, HPV18 and HPV45 co-infections (Table S2). In addition, 396 women in our study had at least one additional serial sample, collected prior to their most recent screening visit ($N = 974$ samples), available for inclusion (Table S3; serial time-point samples). We included all available serial samples. In total, there were 3929 samples collected from 3351 women in the study (Tables S1 and S3).

Because all samples are from a prospective cohort, we conducted two sets of analyses: 'single time-point' analyses, and 'serial time-point' analyses. For 'single time-point' analyses, we included one sample per woman, collected during the antecedent screening visit close to (within 2 years, $N = 3031$) or far from (≥3 years, $N = 320$) their clinical diagnosis date (total $N = 3351$ samples) (Table S1). We categorized these samples based on these two time periods from diagnosis because it is possible that women could have missed or undetected disease during a 2 year time frame prior to diagnosis depending simply on when their screening visits were scheduled. For 'serial time-point'

analyses, we included 2–5 samples per woman, collected up to 10 years before diagnosis ($N = 974$ total samples from 396 women). We categorized 'serial time-point' samples based on their collection time from diagnosis: time-point (TP) one or TP1 = closest to or at the time of diagnosis; TP2 = second available sample collected, next to TP1; TP3 = third available sample collected, next to TP2; TP4 = fourth available sample collected, next to TP3, and TP5 = fifth available sample collected, next to TP4 (Table S3). All case samples were collected before precancer/cancer diagnosis.

We additionally included 32 exfoliated cervical cell samples from women in PaP that were HPV-negative with normal cytology and tested negative for HPV for at least 2 consecutive screening visits, to investigate the occurrence of somatic mutations in HPV-negative cervical cell samples compared to HPV-positive samples.

### DNA extraction and sequencing

Selected samples were transferred to the National Cancer Institute, Cancer Genomics Research (CGR) laboratory. DNA was isolated by transferring 30 μL of the STM specimens to a buffer containing 200 μg/mL of proteinase K, followed by incubation at 55 °C and 95 °C for 2 h and 10 min, respectively[33]. DNA was prepared for sequencing using Thermo Fisher Life Science Ion Torrent S5 GeneStudio platform (Thermo Fisher Scientific, Waltham, MA, USA). A set of custom primers were designed to amplify the exonic region of 20 genes that have been previously described as significantly mutated in ICC[4,5,11], including *ARID1A*, *CASP8*, *ELF1*, *EP300*, *ERBB2*, *ERBB3*, *FBXW7*, *HLA-A*, *HLA-B*, *HRAS*, *KRAS*, *MAPK1*, *MED1*, *NFE2L2*, *PIK3CA*, *PIK3R1*, *PTEN*, *STK11*, *TGFBR2*, *TP53*. Libraries were constructed using AmpliSeq Library Preparation kit 2.0-96LV (Thermo Fisher Scientific, Waltham, MA, USA). Library quantification was performed with the Kapa Biosystems Library Quantification Kit-IonTorrent/LightCycler 480 (Roche, Basel, Switzerland), and Agilent BioAnalyzer DNA High-Sensitivity LabChip (Agilent Technologies, Santa Clara, California). The average read depth per amplicon for all samples was 820x (SD 527), and the average/median coverage metrics for each sample are provided in Supplementary Data 1.

### Mutation calling and quality control

An in-house pipeline was developed to analyze the amplicon panel, detailed in the Supplemental Material. First, sequence reads underwent read quality assessment and trimming, then reads were mapped to the human reference genome hg19 using Torrent Suite software (Thermo Fisher Scientific, Waltham, MA, USA). Somatic variant calling was performed in a single sample fashion without paired normal samples (e.g., tumor-only), given that our samples are cervical cells from residual pap-smears in a clinical setting without a matched blood collection. Single nucleotide variants were called using the Torrent Variant Caller (TVC) v.5.0.3 (Thermo Fisher Scientific, Waltham, MA, USA) with the manufacturer recommendations with low-VAF parameters for a minimum allele fraction of 2%.

To assess the performance of our amplicon-based assay to detect known somatic mutations at a low VAF and to establish filters to clean potential false positive variants, we sequenced the Acrometrix Oncology Hotspot Control DNA (AOH) (Thermo Fisher Scientific, Waltham, MA, USA), detailed in the Supplemental Material: Mutation calling quality control). We applied the following variant calling parameters established in this experiment to improve detection of true low VAF somatic mutations: FILTER = PASS (passed TVC default variant calling parameters), QUAL ≥ 10, FDP ≥ 100, FAO ≥ 6, STB < 0.9, MLLD > 55. We achieved between 70.0% to 98.9% sensitivity for detecting known mutations with an allele fraction between 2% and 35% (Table S4 and Fig. S1). In the serial time-point analyses, we relaxed these filtering criteria in TP2 and beyond to ≥1 read per mutation and ≥100 read depth only for the specific *loci* with mutations detected in TP1.

SnpEff v.3.6c[34] was used to annotate synonymous and non-synonymous mutations, indels, and frameshifts, and Annovar[35] was

                                                  

used to annotate exonic and non-exonic (e.g., introns, UTR) mutations, the gnomAD frequencies, and COSMIC information. To remove potential known germline variants and to keep known somatic mutations, we excluded mutations reported with a minor allele frequency (MAF) ≥ 0.01 in gnomAD[36], and kept only those reported in COSMIC v92[37]. Then, to keep mutations more likely to be cancer drivers, we excluded synonymous mutations and mutations located in intronic regions that did not affect splice sites. Lastly, we excluded mutations located in homopolymers with ≥3 bases and those that mapped to repetitive regions (details described below) (Fig. 1).

## Hotspot mutations and viral genetic variation classifications
To identify somatic hotspot mutations previously described (i.e., not to discover new mutations), we used the following somatic mutation databases for previous cancer genomics studies: the Cancer Genome Interpreter (CGI)[38], Mutagene[39,40], cBioPortal-TCGA-cervix (Cervical Squamous Cell Carcinoma and Endocervical Adenocarcinomas)[4,41,42], and CHASMplus[43]. A detailed description is in Supplemental Material. Based on these databases, we classified previously reported mutations as either known drivers in ICC (i.e., TIER1) or known drivers in other cancers (i.e., TIER2). Specifically, we assigned mutations to TIER1 (restricted classification) if they were mutations reported to be drivers in ICC by CGI or Mutagene, or an amino-acid (aa) change in ≥2 samples from the cBioPortal-TCGA-cervix database; or to TIER2 (expanded classification) if they were mutations reported to be drivers in other cancers by CGI or Mutagene, or mutations at the same aa position in ≥3 samples from the cBioPortal-TCGA-cervix. Therefore, a hotspot mutation is defined as a mutation previously observed in the aforementioned somatic mutation databases and classified as TIER1 or TIER2. Mutations were classified as being potentially induced by APOBEC3 if they were a C > T or C > G DNA change occurring at a 5′TCW3′ [W is A or T] trinucleotide motif[12]. All hotspot mutations were visually inspected by manually reviewing each hotspot nucleotide position in the aligned reads in IGV[44] and excluded if they were present in repetitive regions of the genome, called by ambiguously mapped reads (mapping Quality=0), reads with low base quality (quality ≤20), or showed forward or reverse strand bias[45].

We classified the other non- hotspot mutations as TIER3 (new mutations in our cohort predicted as drivers by CGI and reported as common in CHASMplus) or passengers (known as passengers or likely neutral by CGI or Mutagene). Mutation filters, counts and classification criteria are summarized in Fig. 1. We categorized samples as having no hotspot mutation, at least 1 hotspot mutation or 2 or more hotspot mutations.

To assess hotspot mutations by disease status and HPV type/variant, we combined HPV18 and HPV45 for most statistical analyses due to the limited mutation counts in each stratum by histology, because they are genetically related, both are part of the *Alpha 7* species group, and they have a similar relative higher frequency among ADC compared to SCC. We categorized samples as HPV16-positive, when HPV16 was present as a single infection or with multiple types other than HPV18 or HPV45, and as HPV18/45-positive, when HPV18 and/or HPV45 were present as single or concurrent infections or with multiple types other than HPV16.

## Statistical analyses
To compare hotspot mutation frequencies, we used logistic or multinomial logistic regression models to assess differences by disease status and across histology and HPV types, calculating the odds ratio (OR) and 95% confidence intervals (CI). Disease outcomes were defined as controls (i.e., HPV transient infection: < CIN2 or subsequently cleared infection), CIN2, precancers (CIN3 or AIS), and ICC (including SCC and ADC). When comparing controls versus precancer or cancer cases, we also adjusted models for age, smoking status, and race/ethnicity, and none of these covariates affected the direction and

strength of associations, therefore we are presenting only the unadjusted models. To assess differences in the VAF of hotspot mutations across disease status, we used the non-parametric Mann-Whitney test. We investigated the HPV type/lineage-specific[18,46] (see Supplemental Material: HPV genome sequencing and lineage assignment) association between hotspot mutations and disease status, using a generalized linear mixed-effects model. To assess differences in the allele fraction of hotspot mutations across serial time point samples we calculated a rate of VAF change 'r' by years [formula ($r = $ VAF.TP$_2$ − VAF.TP$_1$ / years.TP$_2$ − years.TP$_1$)] and tested differences with the non-parametric paired Mann-Whitney test. All statistical analyses were performed in R version 4.1.0.

## Reporting summary
Further information on research design is available in the Nature Portfolio Reporting Summary linked to this article.

## Data availability
The somatic gene sequence data generated in this study have been deposited in dpGAP under accession code phs003691. The de-identified phenotype data are available under this accession. Source data are provided with this paper.

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

## Acknowledgements

The content of this publication does not necessarily reflect the views or policies of the Department of Health and Human Services, nor does it mention that trade names, commercial products, or organizations imply endorsement by the U.S. Government. This research was funded by the intramural research program of the Division of Cancer Epidemiology and Genetics, National Cancer Institute, NIH. This project has been funded in whole or in part with federal funds from the National Cancer Institute, NIH (HHSN261200800001E), the National Cancer Institute (CA78527 and CA238592).

## Author contributions

Study conceptualization: M.P. and L.M. Data curation: M.P., L.M., A.S., J.F.B., S.B., M.Y., and S.M. Formal analysis: M.P., B.Z., L.M., and Z.C. Funding acquisition: L.M., M.S., N.W. Methodology: B.Z., L.M., L.B., M.Y., and S.M. Resources: L.M., M.S., N.W., P.E.C., R.D.B., and T.L. Supervision: L.M. and B.Z. Writing the original draft: M.P. and L.M. All authors have read, reviewed, and agreed to the published version of the manuscript.

## Funding

## Competing interests

M.P. is currently an employee of GlaxoSmithKline, J.F.B. and S.B. are now employees of AstraZeneca, and Amulya Shastry is now a doctoral student at Boston University, but they all completed the work associated with this project while employed at the National Cancer Institute. P.E.C. has received HPV tests and assays at a reduced or no cost for research from Roche, Becton Dickinson, Cepheid and Arbor Vita Corporation. All other authors declare no competing interests.
