## [Peer Review File · Nature Communications]

Somatic mutations in 3929 HPV positive cervical cells associated with infection outcome and HPV typeREVIEWER COMMENTS

Reviewer #1 (Remarks to the Author): computational expertise in cervical cancer

The title is suggestive and should be revised. The investigators did not specifically examine whether the mutations are driving carcinogenesis, and the sample size is short of 4,000.

Line 51: Do the authors mean evolve to ICC, or invade the surrounding tissue to become ICC?

Line 73: Which population are the authors referring to? Global or US only?

Line 91: What do the authors mean by benign HPV infections? Clarify, use standard language in the field, and be consistent throughout the manuscript.

Line 90-95: The findings from the present study should be included in the results and discussion sections of the manuscript, not the introduction.

The study population is confusing.

There is no scientific justification for conducting serial time-point analyses, using a control:case ratio of 1.13:1, or adding 32 HPV negative samples. It is unclear if any of the samples included in the analyses had multiple infections with HPV16 and 18/45, and how were those addressed. The precancer with unknown histology and the serial samples should be excluded from the current study. The results of these analyses would be more accurate and credible if the samples are independent – using only the samples collected during the antecedent screening visit (N=3,031), analyses should have mutually exclusive datasets, and results for n=1 should be excluded.

Controls are defined as “women having baseline HPV16-, HPV18- and HPV45-positive specimens and no histologic evidence of equivocal precancer or worse (CIN2+) during the study follow-up period according to data obtained from the electronic health records” and as “A HPV benign infection < CIN2 or cleared infection”. The authors should clarify the definition of the controls and be consistent. Were individuals who cleared HPV infections included as controls in this study? If so, what was the timeline for HPV clearance?

Importantly, the authors suggest very strongly that they detected important somatic driver events before cancer diagnosis. The diagnosis may have been missed; the diagnostic parameters (including personnel) may have changed over time; the timeline between the sample collection and the diagnosis should be accounted for in the analysis. The limitations of this study should be elaborated.

Table s2 should be revised to make the relationship between the timepoints clear. For example, how many years apart are TP1 and TP4?

This reviewer disagrees with some of the novelty claims in this study. The authors should reference similar recently published studies, including Analysis of somatic mutations and key driving factors of cervical cancer progression, Niyazi M. et. al 2023 Open Medicine. Further, the findings in this study should be confirmed, before suggestions about translational utility are made.

Reviewer #2 (Remarks to the Author): computational expertise in cervical cancer

This is great work by authors who are leaders in this field. It has a great study design and efficient use of the PaP study resources, which have accrued for many years. I have only a few suggestions.

Table S4 seems central to the story, and if journal rules permit, it should be brought to the main narrative.

Why didn't the authors include HPV31 and HPV35 in their work? The same rationale applied to studying HPV18 and HPV45 would have been applicable to these strong players in the Alpha 9 species.

Does the PaP cohort have a sizable number of cases driven by 'lesser' carcinogenic HPVs? The HPV science community often wonders if lesions caused by these lesser types would take longer to

develop and thus have more of an ample opportunity to be discovered via screening. Vaccination targets the more prominent HPV types. Would the number of mutations be larger for these other types before irreversible malignancy sets in? This is a relevant question for the post-HPV vaccination era. I wonder if the authors thought about taking advantage of the contrast with less prominent carcinogenic HPV types.

Did the authors examine if some HS mutations were associated with early disease onset? Was data available on disease stage or other clinical features of aggressive disease that could have been used to identify 'bad' HS mutations?

Reviewer #3 (Remarks to the Author): expertise in computational cervical cancer genomics

Pinheiro et al. report on somatic mutations in a large cohort of 3,929 exfoliated cervical samples using a gene panel. They have follow-up samples available for around 10% of their cohort and report on the rate of change of variant allele frequencies for specific hotspot mutations. They also investigate the HPV type specificity of these mutations.

Major comments:

1. The gene panel used in this study consists of genes reported previously in cervical cancer cohorts. While replication studies are necessary and may expand on previous work, could you clarify the following:
ARID1A, CASP8, ELF1, EP300, ERBB2, ERBB3, FBXW7, HLA-A, HLA-B, HRAS, KRAS, MAPK1, MED1, NFE2L2, PIK3CA, PIK3R1, PTEN, STK11, TGFBR2, TP53 were investigated in this study. These are twenty genes, not nineteen (line 151).
Have you investigated ELF1 or ELF3 (as in Ojesina et al. 2013 – ref 6)?
The rationale behind selecting these genes is slightly unclear – the genes PIK3CA, PTEN, TP53, STK11, KRAS, MAPK1, HLA-B, EP300, FBXW7, NFE2L2, ELF3(1?) and ERBB2, were taken from Ojesina et al. 2013, but not CFBF, the genes ERBB3, ARID1A, CASP8, HLA-A and TGFBR2 arise from the TCGA study (2017 – ref 5) but the gene SHKBP1 was not taken. HRAS comes from (Lou et al 2015 – ref 11), but not CTNNB1 or CDKN2A. I could not find MED1 and PIK3R1 in the references provided. Please elaborate.
2. Line 94, exfoliated cervical cells – cells derived from cervical scrapings are a mix of cervical epithelial cells and immune cells, and likely also consist of tumor and normal cervical cells. Are there any known cell surface markers/available gene expression data on this cohort to determine the nature of the cells studied? The summary lines, although cautious, propose the potential use of this method for early detection of cancer driving mutations, however, a matched germline control, as well as confirming the “cervical tumor” origin of the samples might be necessary.
3. Line 112, HPV Other positive and the types are not mentioned here, although they appear later in the manuscript (lines 211-213). Please specify the other types and provide a supplementary table with sample numbers per HPV type, single and double positives, across histology. What about CIN1 samples – why were these excluded? Line 141-143- Were these women also histology negative for lesions? Kindly clarify the controls mentioned on line 219 and on line 248 – are they HPV negative or positive? Are any of these CIN1+? What is meant by benign transient control (line 470)?
4. Line 158 – an average read depth of 820x is provided for all samples and genes. Were all sample-gene combinations covered adequately? Please provide the median coverage (and range), and sequencing statistics of each sample in a Supplementary Table.
Line 161 – Provide the parameters underlying read quality assessment, cutoffs, filters etc. either in the main text or supplementary material with reference.
Same for line 171, specify the filtering criteria or refer the reader to where it can be found. Please specify the values for exclusion in line 201 (reads with low base quality).
5. Lines 176-182 and Figure 1- almost 90% of the discovered mutations were discarded since they were unavailable in COSMIC. Can any further variants be included by comparing with known variants in CGI, cBioportal-CESC, or CHASMplus (lines 186-187)? TIER 1 and TIER 2 classification purely based on the tissue of interest is not ideal - if a cancer-causing mutation has not yet been

reported in cervical tissue, that does not make it any less important as the loss of function effect remains the same. I recommend combining TIER 1 and 2 together, perhaps a gene based analysis would be more meaningful (combining LoF mutants per gene for statistical analysis). See also lines 437-441 and 444-446.

6. Kindly provide a list of the mutations detected per sample over each time point as a supplementary file. In case two variants occur in a woman across two consecutive time points, which mutation appeared first or was the driver – this would provide relevant information such as in the case of TP53 and PIK3CA somatic mutations in endometrial cancer (see PMID 24170611). For samples with hotspot mutations - is it possible that likely relevant variants on other genes from the panel have been excluded due to the COSMIC/novelty filtering?

7. One of the limitations of the study is already noted by the authors in the discussion – the lack of matching normal tissue/blood DNA to filter out true germline variants. I wonder if the 32 HPV negative disease-free cervical cell samples (Lines 254 to 256) could have been used to identify germline mutations specific to the cervix, as an alternative to lines 482-484. (Unfortunately, SomamutDB PMID: 34634815, so far lacks uterine/cervical tissue, otherwise known somatic mutations could have been filtered out as well).

8. In 207-213, it is mentioned that HPV18+ and HPV45+ samples were combined, but in the results (265-266), these have been analysed separately as well. Lines 305-307 mention results from some HPV sub-lineage analysis – please expand on this in the methods section as well.

9. The VAF rate change calculation is quite interesting. For the variants where a rate of VAF change was calculated, do these rates vary between type of mutation (e.g. missense, STOP or frameshift)?

10. In Table 1, why were CIN2, CIN3/AIS groups not shown under N, PIK3CA and N, non-PIK3CA? Were these early or late mutations? Similarly under Table 2 and 3.

Minor comments:

1. Title – Please write the exact number of samples used in the study (3929, not 4000).

2. The references are misplaced – for example, line 152, ref 13 (Evolution and taxonomic classification of alphapapillomavirus 7 complete genomes: HPV18, HPV39, HPV45, HPV59, HPV68 and HPV70) has no connection with the gene panel selected. Sometimes the references are in parentheses, other times as superscript – rechecking required.

3. Line 46, 13 hrHPV types – ref 2 Table 2 lists 15 hrHPV types. Or is this defined by the molecular test used to determine HPV type in the samples?

4. Line 49 “goes from” -> “starts with”, line 51 “invade to ICC” -> “develop into ICC”?

5. Typo – Line 57 NF2L2 instead of NFE2L2.

6. Line 98, what is the criteria for the selection of the women in this study? Please provide age (range), and ethnicity information or any inclusion/exclusion criteria.

7. Line 123 – 124 reads a bit vague, please specify the number: “A subset of the total CIN2 cases available were randomly selected for inclusion”

8. The line 195-196 is repetitive (same as line 184).

9. Line 176 – snpEFF and ANNOVAR were used for annotating mutations – which information was taken?

10. Line 234 – 246 – why are mutations called mutated site (only in this paragraph)?

11. Lines 363-365 rate of VAF change written as $1.32 \times 10^{(-1)}$, perhaps 0.13 or 13% increase would be better? What is meant by “per HPV”?

12. Line 371 and 459 – the 76 fold change is not mentioned in the results previously, although it appears in table 3. Perhaps a mention in the results, along with the other “not so striking”, but still significant fold changes would be appropriate.

13. Figure 2c TIER 2 - p value cannot be 68.

14. Figure 4, please show scatter dot plots, instead of box plots.

RESPONSE TO REVIEWERS' COMMENTS

We appreciate the reviewers' time and thoughtful comments that have improved our revised manuscript. Please find our point-by-point responses to their verbatim comments below in blue.

Reviewer #1 (Remarks to the Author): computational expertise in cervical cancer

The title is suggestive and should be revised. The investigators did not specifically examine whether the mutations are driving carcinogenesis, and the sample size is short of 4,000.

Response 1: Following the reviewer's suggestion, we have revised the title to "Somatic mutations detected in 3,929 HPV-positive exfoliated cervical cell samples are associated with infection outcome and HPV type".

Line 51: Do the authors mean evolve to ICC, or invade the surrounding tissue to become ICC?

Response 2: We meant that the persistent HPV infections that lead to the development of precancerous lesions, which grow within the epithelium, can eventually 'invade the surrounding tissue to become' ICC. We have revised this sentence to clarify this point as suggested.

Line 73: Which population are the authors referring to? Global or US only?

Response 3: This is for a global population. HPV16, HPV18 and HPV45 are the three most common types detected in cervical cancer worldwide. We have added 'worldwide' to that sentence.

Line 91: What do the authors mean by benign HPV infections? Clarify, use standard language in the field, and be consistent throughout the manuscript.

Response 4: We meant the benign HPV infections as the HPV16/18/45 positive infections that either cleared (i.e., HPV was no longer detected at a subsequent screening visit) or the infection never progressed to cervical intraepithelial neoplasia grade 2 (CIN2) or 3 (CIN3) precancer or cancer during the study. Based on the cervical intraepithelial neoplasia (CIN) classification system, it is standard language in the field to note this as '<CIN2'. To be more clear and consistent throughout the manuscript, we now write 'transient' HPV infection instead of 'benign' when referring to our controls, we have added additional details to the study population description, and added '<CIN2 or subsequently cleared infection' in parentheses with the mention of transient HPV infections.

Line 90-95: The findings from the present study should be included in the results and discussion sections of the manuscript, not the introduction.

Response 5: We have revised these introduction sentences to only state the analyses planned and not the results, as suggested.

The study population is confusing.

There is no scientific justification for conducting serial time-point analyses, using a control:case ratio of

1.13:1, or adding 32 HPV negative samples. It is unclear if any of the samples included in the analyses had multiple infections with HPV16 and 18/45, and how were those addressed. The precancer with unknown histology and the serial samples should be excluded from the current study. The results of these analyses would be more accurate and credible if the samples are independent – using only the samples collected during the antecedent screening visit (N=3,031), analyses should have mutually exclusive datasets, and results for n=1 should be excluded.

Response 6: We apologize that our description of the study population was confusing, we have added more details to the study population descriptions. Progression to precancer (CIN3/AIS) and cancer is a rare HPV infection outcome, therefore, we selected all the available HPV16-, HPV18- and HPV45-positive precancer and cancer cases that had samples available from our NCI-Kaiser PaP cohort. We then randomly selected HPV16-, HPV18- and HPV45-positive controls at a 1:1 ratio to the precancer/cancer cases, and also added some extra control samples to be sure we reached this ratio accounting for potential methodological and laboratory failures. We had very few failures so we ended up with a control:case ratio of 1.13:1. We think it is best to include all the controls that were sequenced instead of restricting to a 1:1 ratio.

We included 32 HPV-negative samples, which were samples that tested negative for any HPV for at least 2 consecutive screening visits, to evaluate if we detected somatic mutations in the HPV-negative cervical cell samples for comparison to our HPV-positive samples. In addition, this analysis was valuable to inform whether the somatic mutations observed in these cells resulted from cervical tissue background mutations (i.e., noise). Since our study predominantly sampled from HPV-positive women, we did not have many women that tested negative for any HPV type for 2+ consecutive screening visits and also had available cervical cell samples for analysis. Therefore, we tested all that were available (n=32).

There was one precancer with unknown histology, and 11 cancer cases with unknown histology, this is noted in the study population description. We included all of these cases in our combined precancer and cancer analyses that are not stratified by histology to optimize our power. We also tested the ‘all cancer’ (not stratified by histology) analyses after excluding these 11 cancers and the results did not differ from the results when including these 11 cancers. We aren’t aware of a reason to exclude them from the analyses of precancers and cancers, these aren’t questionable cases, they are only unknown in terms of squamous or glandular histology. Therefore, to utilize all the available cases for each analysis, we include these cases for the analyses that are not stratified by histology.

For multiple infections, among our 3,351 single time-point samples, 136 samples had HPV18 and HPV16 co-infections, 21 had HPV18 and HPV45 co-infections, 84 had HPV45 and HPV16 coinfections, and 9 had HPV16 and HPV18 and HPV45. These co-infected samples were only excluded in the single HPV type stratified analyses. The HPV co-infection counts were added to the study population description (lines 130-132), we also added a new supplementary Table S2 that provides the number of women with a single type infection and those with each coinfection, and the following was added to the results text:

“First, we evaluated the distribution of hotspot mutations among single HPV16, HPV18, and HPV45 infections only (i.e., HPV co-infected samples were excluded).” (lines 280-281)

For the association analyses by disease status/histology and HPV type/variant, we combined HPV18 and HPV45 infections due to the limited number of mutation counts in each stratum by histology because HPV18/45 are genetically related, both are part of the Alpha 7 species group, and they have a similar relative higher frequency among ADC compared to SCC. We categorized samples as HPV16-positive, when HPV16 was present as a single infection or with multiple types other than HPV18 or HPV45, and as HPV18/45-positive, when HPV18 and/or HPV45 were present as single or concurrent infections or with multiple types other than HPV16. This is described in the methods on lines 227-233.

The vast majority of our analyses were focused on only the antecedent screening visit samples within a 2-year interval from diagnosis (N=3,031), which included only single time-point independent samples that were mutually exclusive. Serial samples were only included in the analyses where specifically noted, in Supplemental Figures S4B and S5, Figure 4b, and Tables S4 and S11, and only described in the last section “Allele fractions of hotspot mutations are highest in ICC and increase over time” of the results. We think the serial samples are a very valuable addition to the antecedent screening visit samples, as it is rare to have this many serial case samples from a prospective cohort and they provide unique insights into mutation allele fraction changes and detection prior to clinical diagnosis. We agree results based on only n=1 sample should be interpreted with caution; however, this is the first time this has been evaluated, and we include all samples because detecting even one rare TIER1 hotspot mutation in a single sample collected prior to clinical diagnosis is novel, rare, and could be potentially important. Instead of excluding the n=1 samples, we have highlighted the row in Table 2 where the total sample count is only n=1 and added a footnote to note that these results should be interpreted with caution. We do not focus our conclusions on results based on one sample, we instead focus on the statistically significant differences that we observed.

Controls are defined as “women having baseline HPV16-, HPV18- and HPV45-positive specimens and no histologic evidence of equivocal precancer or worse (CIN2+) during the study follow-up period according to data obtained from the electronic health records” and as “A HPV benign infection < CIN2 or cleared infection”. The authors should clarify the definition of the controls and be consistent. Were individuals who cleared HPV infections included as controls in this study? If so, what was the timeline for HPV clearance?

Response 7: We have revised the study population description and the text to be more consistent. Please also see response #4. These two control definitions quoted are the same, the latter is the shortened version of the former definition. An HPV16-, HPV18- and HPV45-positive infection and no evidence of CIN2+ during the study follow-up period, in the first definition noted by the reviewer, includes all infections less than CIN2 including those that subsequently cleared, as noted in the later definition as HPV benign infections < CIN2 or cleared infections. In the revised methods, controls are defined as follows:

“Controls were defined as women having baseline HPV16-, HPV18- and/or HPV45-positive specimens that subsequently cleared their infection and/or had an infection defined as normal or low-grade lesion (ASCUS, LSIL, CIN1) with no histologic evidence of equivocal precancer or worse (CIN2+) during the study follow-up period according to data obtained from the electronic health records.” (lines 121-125)

The most typical and common HPV infection outcome is clearance, this occurs in approximately 80% of women within the first three years of infection (Demarco et al., EClinicalMedicine 2020:22:100293). For our controls, 77.4% subsequently cleared their infections during follow-up with a timeline of a few months to several years. However, timing is based on the time of the routine screening visits and given the screening visit intervals are fixed and variable among women, the actual HPV clearance date could have happened prior to the subsequent visit timing; therefore, a precise estimate of clearance time is difficult to estimate and instead of using an exact estimate based on screening visit dates we use a time interval or time-frame. We added this clearance information to the methods for the controls that cleared their infections (lines 125-126). Please also see response #8.

Importantly, the authors suggest very strongly that they detected important somatic driver events before cancer diagnosis. The diagnosis may have been missed; the diagnostic parameters (including personnel) may have changed over time; the timeline between the sample collection and the diagnosis should be accounted for in the analysis. The limitations of this study should be elaborated.

Response 8: We agree with the reviewer, the disease may have been present prior to the date of a woman's screening visit where the clinical diagnosis was made. Although this is a prospective cohort and all of the included samples were collected prior to the clinical diagnosis, the sample collection time before clinical diagnosis varies. We accounted for this in our analysis by grouping the woman based on the time interval between the sample collection date and the clinical diagnosis date; and we categorized the samples that were collected within 2-years of the date of diagnosis as 'at diagnosis' samples. This allowed us to account for potential missed diagnosis or presence of undetected disease during a 2-year time frame from the scheduled screening visit. While we think this is a good approach, it is possible we are including some samples that were actually collected prior to disease as 'at diagnosis', and conversely, it is also possible that a longer than 2-year time interval underlies the prevalent disease. This approach is described in the methods at lines 138-144, and we have expanded our limitations to make this clear as a limitation.

One of the benefits of working with KPNC is the consistency of the diagnostic parameters and routine screening over the study period, so we don't think this is an issue with this cohort.

Our study limitations are discussed in the discussion starting at line 515. We have added the following as an additional limitation of our study as the reviewer suggested:

"Our prospective cohort includes cervical cell samples collected from women undergoing routine cervical cancer screening, therefore, the precise timing of disease diagnosis may be limited by the timing of the screening visit intervals. To account for potential undetected disease due to the screening visit intervals, we grouped the samples collected within 2-years of the date of clinical diagnosis as 'at diagnosis' samples. However, we cannot exclude that a longer time interval underlies prevalent disease."

Table s2 should be revised to make the relationship between the timepoints clear. For example, how many years apart are TP1 and TP4?

Response 9: We have added a footnote to Table S3 (formerly Table S2) to describe the relationship between the timepoints. These serial time-point categories are based on the delta values between the sequential time-points, because the timing between samples varies, see responses #7 and 8, we categorized the serial time-point samples based on the time-frame from diagnosis or the preceding serial sample. We show the counts for these time-frames for each serial sample time-point and the maximum year between the samples in that group. For example, in Table S2, the TP1 columns give the counts for the number of samples with ≤ 2 years or ≥ 3 years time-frame difference between the most recent and the second most recent serial sample, and the maximum number of years between them.

This reviewer disagrees with some of the novelty claims in this study. The authors should reference similar recently published studies, including Analysis of somatic mutations and key driving factors of cervical cancer progression, Niyazi M. et. al 2023 Open Medicine. Further, the findings in this study should be confirmed, before suggestions about translational utility are made.

Response 10: We thank the reviewer for pointing out this study, Niyazi et al 2023 Open Med (Wars), we had not seen it and apologize for not referencing it. This is a nice study looking at somatic mutations using WES in 52 paired cervical tumor tissue and peripheral blood samples. This study is similar in that they also observe differences in somatic SNV mutation frequencies between LSIL lesions, HSIL and cancers, with the highest frequency in cancers, and they highlight that APOBEC mutagenesis is a risk factor for cervical cancer and may cause somatic mutations. Although the mutated genes they highlight, to our knowledge, are not known cervical cancer driver genes in the literature (our TIER1 genes) and they do not overlap with the genes we are highlighting. We have added reference to this suggested study.

Our study is novel for several reasons: (1) large set of samples collected prior to clinical diagnosis at routine cervical cancer screening visits, (2) we are using serial cervical cell samples from a prospective study to observe changes in mutation frequencies in samples collected in current clinical practice, and (3) we are evaluating somatic mutations by HPV type/group and genetic variants. To our knowledge, we are the first study to detect and characterize cervical cancer somatic driver mutations in exfoliated cervical cell samples from routine cervical cancer screening, as compared to tumor blocks or tissue biopsies; and, the first study to demonstrate that HPV type/group/variants influence the somatic landscape. We have revised the discussion to highlight the specific novelty of our study more clearly, and we have revised our conclusions to state that our findings should be confirmed (see line 540), as suggested.

Reviewer #2 (Remarks to the Author): computational expertise in cervical cancer

This is great work by authors who are leaders in this field. It has a great study design and efficient use of the PaP study resources, which have accrued for many years. I have only a few suggestions.

Thank you.

Table S4 seems central to the story, and if journal rules permit, it should be brought to the main narrative.

Response 1: We agree that the data in Table S4 is central to the story, since the same TIER1 and TIER2 data were illustrated in Figure 2A we had included this table as a supplement, but we have now moved it to the main narrative as the new Table 1, as suggested.

Why didn't the authors include HPV31 and HPV35 in their work? The same rationale applied to studying HPV18 and HPV45 would have been applicable to these strong players in the Alpha 9 species.

Response 2: We focused our study on a thorough evaluation of the three most common carcinogenic types, HPV16, HPV18 and HPV45, and use all the available case samples for these types, with a matched number of controls. We thought it was most important to maximize our power to detect differences among our strata by using all the available case samples for these types. Specifically, this allowed us to maximize the number of HPV16+ glandular lesions for the Alpha-9 group, which were rarely detected in our HPV31+ and HPV35+ women, for comparisons to HPV18 and HPV45. We would have had to limit the total case counts for each type in order to also include HPV31+ and HPV35+ cases, and adequate controls for comparison, with our study budget. Therefore, we chose to utilize all available HPV16/18/45 case and serial samples, and a matched control set, to enable the most comprehensive analysis of these three most important HPV types by lesion histology and variant. In follow-up to our study, an evaluation of somatic mutations in HPV31+ and HPV35+ infections, and the other less prevalent HR-HPV types, would be valuable for comparison to our data, and we have added this to the discussion (see lines 425-427, and response #3 below).

Does the PaP cohort have a sizable number of cases driven by 'lesser' carcinogenic HPVs? The HPV science community often wonders if lesions caused by these lesser types would take longer to develop and thus have more of an ample opportunity to be discovered via screening. Vaccination targets the more prominent HPV types. Would the number of mutations be larger for these other types before irreversible malignancy sets in? This is a relevant question for the post-HPV vaccination era. I wonder if the authors thought about taking advantage of the contrast with less prominent carcinogenic HPV types.

Response 3: This is a great question and comment. The KPNC PaP cohort does have a sizable number of cases driven by the less common carcinogenic types; this cohort has >100,000 samples from approximately 45,000 HR-HPV+ women, which allows us to capture the more rare HPV infections and outcomes. However, the numbers are still relatively small compared to what we have included here for our current study of HPV16/18/45. Because our study has identified that HPV18 and HPV45 have more mutations than HPV16, we agree it's possible that the less common carcinogenic types may have an even greater number of mutations. We hope to conduct a follow-up study to characterize and compare somatic mutations in the lesser carcinogenic types. We have added this suggestion to the discussion at lines 424-425.

Did the authors examine if some HS mutations were associated with early disease onset? Was data available on disease stage or other clinical features of aggressive disease that could have been used to identify 'bad' HS mutations?

Response 4: We did evaluate if HS mutations were associated with age at precancer/cancer diagnosis and we did not identify any HS mutations significantly associated with earlier age of disease onset (i.e., earlier precancer and/or cancer diagnosis age). We also conducted adjusted models for age, smoking

status, and race/ethnicity, when comparing the controls vs. precancer and cancer cases, and none of these covariates affected the direction or strength of the associations. Unfortunately, we do not have disease stage or other clinical features of aggressive disease for this cohort. This is in the methods at lines 239-242.

Reviewer #3 (Remarks to the Author): expertise in computational cervical cancer genomics

Pinheiro et al. report on somatic mutations in a large cohort of 3,929 exfoliated cervical samples using a gene panel. They have follow-up samples available for around 10% of their cohort and report on the rate of change of variant allele frequencies for specific hotspot mutations. They also investigate the HPV type specificity of these mutations.

Major comments:

1. The gene panel used in this study consists of genes reported previously in cervical cancer cohorts. While replication studies are necessary and may expand on previous work, could you clarify the following:

ARID1A, CASP8, ELF1, EP300, ERBB2, ERBB3, FBXW7, HLA-A, HLA-B, HRAS, KRAS, MAPK1, MED1, NFE2L2, PIK3CA, PIK3R1, PTEN, STK11, TGFBR2, TP53 were investigated in this study. These are twenty genes, not nineteen (line 151).

Have you investigated ELF1 or ELF3 (as in Ojesina et al. 2013 – ref 6)?

The rationale behind selecting these genes is slightly unclear – the genes PIK3CA, PTEN, TP53, STK11, KRAS, MAPK1, HLA-B, EP300, FBXW7, NFE2L2, ELF3(1?) and ERBB2, were taken from Ojesina et al. 2013, but not CFBF, the genes ERBB3, ARID1A, CASP8, HLA-A and TGFBR2 arise from the TCGA study (2017 – ref 5) but the gene SHKBP1 was not taken. HRAS comes from (Lou et al 2015 – ref 11), but not CTNNB1 or CDKN2A. I could not find MED1 and PIK3R1 in the references provided. Please elaborate.

Response 1: We apologize for the typo on line 151, this was corrected from 19 to 20 genes. We have included ELF1 in our study, but we had not included ELF3, CFBF, SHKBP1, CTNNB1 or CDKN2A. Our gene list was selected as a subset of the top genes from the literature. Instead of expanding our list to include more genes, we wanted to instead maximize our sequencing read depth for a select set of genes that were potentially important; we targeted a read depth of >500x per amplicon to improve our ability to detect mutations in our exfoliated cervical cell samples. We agree more genes are important and could be evaluated in a follow-up study. PIK3R1 is somatically mutated at a high frequency in endometrial cancers, and has been detected in routinely collected screening pap samples, which is clinically relevant for early detection (reference: Urlick and Bell, Nat Rev Cancer 2019), and this is similar to our cervical cell specimens from routine pap-smears. MED1 was initially a candidate driver gene for adenocarcinoma's but was later downgraded in importance after our panel was designed. We decided to keep it in the analysis as exploratory, and we only identified passenger mutations in MED1 (new Table 1).

2. Line 94, exfoliated cervical cells – cells derived from cervical scrapings are a mix of cervical epithelial cells and immune cells, and likely also consist of tumor and normal cervical cells. Are there any known cell surface markers/available gene expression data on this cohort to determine the nature of the cells studied? The summary lines, although cautious, propose the potential use of this method for early detection of cancer driving mutations, however, a matched germline control, as well as confirming the “cervical tumor” origin of the samples might be necessary.

Response 2: We agree this would be useful, but unfortunately, there are no available cell surface marker or gene expression data on this cohort to determine the nature of the cells studied, and we don't have RNA prepared from these samples to further evaluate expression. These are residual lysed cervical cell specimens from pap-smears, we have utilized them for HPV typing, HPV-type specific genome sequencing, and this targeted somatic gene sequencing panel. Instead, we utilized a large number of transient infection control samples as a comparison to our precancer/cancer samples and focused on significant differences among these samples for previously reported cervical cancer driver mutations. A study with matched germline samples and that can provide details about the cervical tumor origin of exfoliated cervical cell samples is a worthwhile follow-up. Since we don't have RNA, tumor samples or blood samples from the women in this cohort, it is not the right cohort for that. Although our cohort has many advantages, including the large number of samples collected prospectively and serially from women undergoing routine cervical cancer screening, this is a limitation of the collection. This is noted as a limitation in the discussion, and we suggest follow-up studies with matched germline controls and for detailed evaluations of the origin of the samples (see lines 525-527).

3. Line 112, HPV Other positive and the types are not mentioned here, although they appear later in the manuscript (lines 211-213). Please specify the other types and provide a supplementary table with sample numbers per HPV type, single and double positives, across histology. What about CIN1 samples – why were these excluded? Line 141-143- Were these women also histology negative for lesions? Kindly clarify the controls mentioned on line 219 and on line 248 – are they HPV negative or positive? Are any of these CIN1+? What is meant by benign transient control (line 470)?

Response 3: We apologize for the confusion with our terminology. We have added a new Supplementary Table S2 that provides the number of women with a single type infection, HPV16/18/45 coinfections, and other HR-HPV type coinfections for each histology. There were 136 women that had HPV18 and HPV16 co-infections, 21 had HPV18 and HPV45 co-infections, 84 had HPV45 and HPV16, and 9 had HPV16, HPV18 and HPV45 co-infections. This was added to the methods at lines 130-132.

CIN1 is included as part of our control group. Controls in our study, were women with a benign or transient infection defined as women having baseline HPV16-, HPV18- and/or HPV45-positive specimens that subsequently cleared their infection and/or had an infection defined as normal or low-grade lesion (ASCUS, LSIL, CIN1) with no histologic evidence of equivocal precancer or cancer (CIN2, CIN3, AIS, cancer) during the study follow-up period. CIN1 is considered a low-grade lesion and a sign of HPV infection, along with ASCUS and LSIL. 77.4% of our controls subsequently cleared their HPV infections at later screening visits, which is the most common HPV infection outcome and consistent with the expectation that approximately 80% of women clear their HPV infections in 3 years (Demarco et al., EClinicalMedicine 2020;22:100293). We have added additional details about the study population and

control group to the methods (lines 121-125), added details about the controls to the footnote of Table S1, and revised the language to be more consistent throughout the text.

At the lines 141-143 noted, these are an additional set of samples we included for comparison, these are HPV-negative for all HPV types at 2 or more consecutive screening visits, these were included as a comparison to our HPV-positive samples, to investigate the occurrence of somatic mutations in HPV-negative cervical cell samples. They all had normal or negative cytology/histology. This was added to the methods at line 152. This set of 32 HPV-negative samples was not included as part of our 'control' group, these were only used for comparison to the HPV-positive controls and cases.

The controls mentioned on lines 219, 248, and 470, are all HPV-positive and referring to the same set of controls. We only have one control set, defined as noted above, we apologize for the confusion with our terminology when referring to this group, we have revised the text to only refer to them as 'controls' or 'controls (i.e., HPV transient infection: <CIN2 or subsequently cleared infection)' to be more clear and consistent.

4. Line 158 – an average read depth of 820x is provided for all samples and genes. Were all sample-gene combinations covered adequately? Please provide the median coverage (and range), and sequencing statistics of each sample in a Supplementary Table.

Response 4: The average depth (number of reads) per amplicon per sample was 820x, with a standard deviation of 527. Yes, we have provided a supplementary table with coverage metrics per sample, including average number of reads per amplicon, standard deviation, median, and total coverage per sample (new Supplementary Table S4).

Line 161 – Provide the parameters underlying read quality assessment, cutoffs, filters etc. either in the main text or supplementary material with reference.

Response 5: We have added the parameters underlying read quality assessment, cutoffs, filters, ect, to the Supplemental Material as suggested, and added in the main methods section to refer there for those details, as follows:

“Amplicon panel sequence reads were mapped to hg19 using Torrent Mapping Alignment Program (TMAP; Thermo Fisher Scientific) with the following options: max-adapter-bases-for-soft-clipping = 25; end-repair = 15; min-al-len = 50. Only reads with a mapping quality of ≥ 4 were considered for variant calling. TVC v.5.0.3 (Thermo Fisher Scientific) was used for variant calling with the following parameters: snp_min_coverage = 100; snp_min_cov_each_strand = 4; snp_min_variant_score = 6; snp_min_allele_freq = 0.02; snp_strand_bias = 0.95; snp_strand_bias_pval = 0.01.”

Same for line 171, specify the filtering criteria or refer the reader to where it can be found. Please specify the values for exclusion in line 201 (reads with low base quality).

Response 6: For variant calling parameters, we used TVC v.5.0.3 manufacturer recommendations with low-*VAF* parameters and a minimum *VAF* of 2% to detect somatic mutations (lines 179-181). In addition to the default parameters, we fine-tuned the variant calling in order to improve the calling of the low

allele fraction mutations, we thought this was important given our samples were exfoliated cervical cell samples. These are cells shed from the cervical epithelium, and we expected a lower concentration of precancerous or cancerous cells with somatic mutations, mixed with normal cells. Therefore, we thought another filtering step was necessary to optimize our targeted sequencing panel quality control filters for low VAF mutations, and for that, we used the control panel of known somatic mutations Acrometrix Oncology Hotspot Control DNA (AOH) (Thermo Fisher Scientific). Based on this control panel evaluation to improve the detection of true somatic mutations at a low VAF, we adjusted the variant calling parameters as follows: FILTER = PASS (passed TVC default variant calling parameters), QUAL \geq 10, FDP \geq 100, FAO \geq 6, STB $<$ 0.9, MLLD $>$ 55. With these parameters we achieved between 70.0% to 98.9% sensitivity for detecting known mutations with an allele fraction between 2% and 35% for the control panel. The details of this evaluation and these parameters are described in the Supplemental Methods: Mutation calling quality control section. We have also added these specific parameters to the main text at lines 185-189.

We have added the specific values for exclusion, and added a reference for IGV variant review, as suggested, formerly on line 201, now starting on line 217 as follows:

“All hotspot mutations were visually inspected by manually reviewing each hotspot nucleotide position in the aligned reads in IGV³³ and excluded if they were present in repetitive regions of the genome, called by ambiguously mapped reads (mapping quality=0), reads with low base quality (quality \leq 20), or showed forward or reverse strand bias.³⁴”

5. Lines 176-182 and Figure 1- almost 90% of the discovered mutations were discarded since they were unavailable in COSMIC. Can any further variants be included by comparing with known variants in CGI, cBioportal-CESC, or CHASMplus (lines 186-187)? TIER 1 and TIER 2 classification purely based on the tissue of interest is not ideal - if a cancer-causing mutation has not yet been reported in cervical tissue, that does not make it any less important as the loss of function effect remains the same. I recommend combining TIER 1 and 2 together, perhaps a gene based analysis would be more meaningful (combining LoF mutants per gene for statistical analysis). See also lines 437-441 and 444-446.

Response 7: *We agree that other LOF mutations not previously reported in cervical cancer could be important for cervical cancer, but unfortunately our cervical cell samples here are not suitable for discovery of new mutations. As outlined in Figure 1, we focused on 4,985 rare mutations reported in COSMICv92, which are the somatic mutations previously reported for any cancer (not just cervical cancer). This did remove ~85% of the variants detected, but since our samples are an admixed population of exfoliated cells, and there are background mutations that could only be adequately cleaned to reduce false positives by comparing with matched normal tissue, which we do not have, we think this is the best approach for our study. CGI, cBioPortal-CESC and CHASMplus are only databases for classifying those very same somatic mutations, and using these, we grouped mutations into meaningful TIERS. We specifically evaluated three sets of potentially important somatic driver mutations and also passenger mutations: TIER1 included the known driver mutations in cervical cancers; TIER2 included known somatic mutations reported as drivers in other cancers; TIER3 included the mutations that were predicted as drivers and reported as common, these were mutations that could have represented new somatic mutations; and, passengers included the somatic mutations reported as passengers or likely neutral. The TIER2, TIER3 and passenger mutations were not differentially distributed, or enriched, in*

our cases compared to controls; only the TIER1 known hotspot mutations in cervical cancers were distinctly distributed across disease status groups and enriched in our cases. We think this is an important observation that the TIER2 mutations were not different among our cases and controls (see Figure 2B). Thus, combining TIER1 and TIER2 would mask or dilute the strong associations we observed specifically for TIER1 mutations only. We think the TIER classification is a strength of our study, as described in the discussion. We also note in the discussion as a limitation, that our study was not designed for discovery of novel somatic driver mutations, and thus we limited our analyses to known somatic driver mutations, see lines 523-525. We agree, it is possible that some LOF TIER2 mutations are still contributing to cervical carcinogenesis as secondary hits or they are passengers carried along with pre-malignant cells and they may contribute to genomic instability; this is noted in the discussion at lines 433 and 469-473.

6. Kindly provide a list of the mutations detected per sample over each time point as a supplementary file. In case two variants occur in a woman across two consecutive time points, which mutation appeared first or was the driver – this would provide relevant information such as in the case of TP53 and PIK3CA somatic mutations in endometrial cancer (see PMID 24170611).

For samples with hotspot mutations - is it possible that likely relevant variants on other genes from the panel have been excluded due to the COSMIC/novelty filtering?

Response 8: The TIER1 and TIER2 mutations detected per sample over each time point, and the allele fractions, for the serial samples for each woman are given in supplemental Figure S5. We have also added a new supplementary Table S6 that summaries all the 3,192 mutated sites evaluated in our study. It is possible that variants that are not in COSMIC could be relevant and excluded with our COSMIC filter, however our samples are not suitable for discovery of novel mutations, please see our response 7 above; this is noted as a limitation.

7. One of the limitations of the study is already noted by the authors in the discussion – the lack of matching normal tissue/blood DNA to filter out true germline variants. I wonder if the 32 HPV negative disease-free cervical cell samples (Lines 254 to 256) could have been used to identify germline mutations specific to the cervix, as an alternative to lines 482-484. (Unfortunately, SomamutDB PMID: 34634815, so far lacks uterine/cervical tissue, otherwise known somatic mutations could have been filtered out as well).

Response 9: The reviewer makes a good point, and we did evaluate if we could use the 32 HPV-negative samples as controls, however, we detected very few mutations that were in common with our HPV-positive samples that could be filtered out. There were no overlapping TIER1 and TIER2 mutations.

We agree, it's disappointing that SomamutDB does not include cervical tissue.

8. In 207-213, it is mentioned that HPV18+ and HPV45+ samples were combined, but in the results (265-266), these have been analysed separately as well. Lines 305-307 mention results from some HPV sub-lineage analysis – please expand on this in the methods section as well.

Response 10: We apologize for the confusion; we have revised the methods and results to be more clear about the co-infections. For multiple infections, among our 3,351 single time-point samples, 136 samples

had HPV18 and HPV16 co-infections, 21 had HPV18 and HPV45 co-infections, 84 had HPV45 and HPV16 coinfections, and 9 had HPV16 and HPV18 and HPV45 (see new supplementary Table S2). These co-infected samples were only excluded in the single HPV type stratified analyses, this was revised to the results more clearly (lines 280-281). For the association analyses by disease status/histology and HPV type/variant, we combined HPV18 and HPV45 infections due to the limited number of mutation counts in each stratum by histology because HPV18/45 are genetically related, both are part of the Alpha 7 species group, and they have a similar relative higher frequency among ADC compared to SCC. We categorized samples as HPV16-positive, when HPV16 was present as a single infection or with multiple types other than HPV18 or HPV45, and as HPV18/45-positive, when HPV18 and/or HPV45 were present as single or concurrent infections or with multiple types other than HPV16. This is described in the methods on lines 227-233.

We have added a section to the Supplemental Material to describe the methods of HPV genome sequencing, HPV lineage/sublineage assignment and analyses. Briefly, each single time-point sample had HPV16/18/45 genome sequencing performed, and the viral genome sequences were used for type-specific lineage and sublineage assignments. HPV lineages/sublineages were assigned for each HPV type per woman based on a maximum likelihood phylogenetic tree topology constructed with RAxML MPI (reference doi:10.1093/bioinformatics/btl446) and type-specific HPV16, HPV18 and HPV45 genome sequence FASTA files, including known lineage/sublineage reference sequences for each type. HPV16 sequences were classified as one of the following known sublineages A1-A4, B, C, or D1-D4; HPV18 sequences as a sublineage A1-A5, or B; and, HPV45 sequences as a sublineage A1, A2, B1, B2, or C1. We and others have previously shown that these sublineages have very different associations with disease risk and prevalence in specific human populations. Therefore, we utilize these sublineages as finer within-type HPV categories for comparisons with somatic mutations; and find that only the HPV16 sublineages previously linked to more precancer/cancer, compared to the other sublineages, were more likely to be cancers with TIER1 hotspot mutations.

9. The VAF rate change calculation is quite interesting. For the variants where a rate of VAF change was calculated, do these rates vary between type of mutation (e.g. missense, STOP or frameshift)?

Response 11: We appreciate the reviewer's insightful comments. The rate of VAF change did not significantly vary by mutation type. In the text, we highlight six mutations characterized by the fastest rates of change, including PIK3CA E542K, PIK3CA E545K, ERBB2 R678Q, TP53 R213X, ARID1A R1446X, and PTEN Q17X, ranked by decreasing rates of change. Notably, the top three mutations — PIK3CA E542K, PIK3CA E545K, and ERBB2 R678Q — were all missense mutations in their respective oncogenes. And, the remaining three — TP53 R213X, ARID1A R1446X, and PTEN Q17X — were nonsense mutations in tumor suppressor genes, each resulting in a premature stop codon that leads to the truncation of the respective proteins. This was added to the results at lines 390-392.

10. In Table 1, why were CIN2, CIN3/AIS groups not shown under N, PIK3CA and N, non-PIK3CA? Were these early or late mutations? Similarly under Table 2 and 3.

Response 12: For Table 1, we have added the data for CIN2, CIN3, AIS groups for PIK3CA and non-PIK3CA mutations (see revised Table 2). We were trying to keep the table at a reasonable size and focus on the main cancer results. CIN2 were not significantly different from the controls in any of these noted

analyses. For the CIN3 and AIS precancers, there were significantly more PIK3CA mutations in these precancers compared with controls, while for non-PIK3CA mutation, CIN3 were comparable to the controls. For Table 2 (comparisons of 1 HS and ≥ 2 HS mutations), there were no significant differences among the CIN2 or CIN3-AIS cases; we believe CIN2 and CIN3/AIS precancers are still earlier stages and did not accumulate 2+ mutations enough for us to be able to detect a difference, so here we focused on differences among cancers only. Table 3 of the comparisons between PIK3CA vs non-PIK3CA mutations by HPV type/group, the numbers were very small and there were no significant differences for CIN2, CIN3, AIS precancers so we left those out of the table and focused on the cancers.

Minor comments:

1. Title – Please write the exact number of samples used in the study (3929, not 4000).

Response: The title was revised as suggested with the exact number of samples.

2. The references are misplaced – for example, line 152, ref 13 (Evolution and taxonomic classification of alphapapillomavirus 7 complete genomes: HPV18, HPV39, HPV45, HPV59, HPV68 and HPV70) has no connection with the gene panel selected. Sometimes the references are in parentheses, other times as superscript – rechecking required.

Response: We apologize for the misplaced references, they have been revised and rechecked.

3. Line 46, 13 hrHPV types – ref 2 Table 2 lists 15 hrHPV types. Or is this defined by the molecular test used to determine HPV type in the samples

Response: There are 13 HPV types that are considered high-risk HPV types. This includes 12 HPV types that are established carcinogens by IARC: HPV16, HPV31, HPV33, HPV35, HPV52, HPV18, HPV39, HPV45, HPV59, HPV51 and HPV56; and HPV68 which is as a probable carcinogen. There are several other types, as noted in ref 2, that are considered possibly carcinogenic in extremely rare circumstances. We have changed that reference (ref 2) to a more updated review of Carcinogenic HPV infections (<https://doi.org/10.1038/nrdp.2016.86>) and it has a better explanation of these types.

4. Line 49 “goes from” -> “starts with”, line 51 “invade to ICC” -> “develop into ICC”?

Response: we have revised these sections noted.

5. Typo – Line 57 NF2L2 instead of NFE2L2.

Response: we have made this change.

6. Line 98, what is the criteria for the selection of the women in this study? Please provide age (range), and ethnicity information or any inclusion/exclusion criteria.

Response: Line 98 and the following paragraph, describes the Kaiser Permanente Northern California (KPNC)-NCI HPV Persistence and Progression (PaP) cohort. We describe the women selected for our study starting at line 112 (formerly line 111). We included all available HPV16-, HPV18- and HPV45-

positive precancer and cancer samples. These cases were defined as cervical intraepithelial neoplasia grade 3, adenocarcinoma in situ, and cancers (denoted CIN3/AIS+). We then included approximately 1 control per case sample (there were a few extra controls to account for potential failures; ratio 1.13:1 case). Controls were defined as women having baseline HPV16-, HPV18- and/or HPV45-positive specimens that subsequently cleared their infection and/or had a benign infection defined as normal or low-grade lesion (ASCUS, LSIL, CIN1) with no histologic evidence of equivocal precancer or worse (CIN2+) during the study follow-up period. A subset of the total CIN2 cases available per type were randomly selected for inclusion. The average age of the women in our study was 38 years (SD 11; ranging from 18 to 88), see the table below by infection outcome for all the women. The majority of the women included in our study self-reported their race/ethnicity as White (51%), followed by Hispanic (20%), Asian/PI (15%), Black (8%), or multiracial/other (7%). This section of the methods was revised to include this suggested information and more details.

Status	Mean AGE	StdDev
Cancers	44.8	11.7
CIN3-AIS	36.9	9.9
CIN2	36.2	11.2
Controls	39.4	11.3
All 3,351 women	38.3	11.0

7. Line 123 – 124 reads a bit vague, please specify the number: “A subset of the total CIN2 cases available were randomly selected for inclusion”

Response: CIN2 is an ambiguous squamous precancer, it is a more common infection outcome compared to the CIN3/AIS precancers, and they are thought to more often regress, so we only selected a random subset of these ambiguous cases for each type for comparisons to the controls and true precancers (CIN3/AIS). The specific counts by HPV type are given in a new Table S2.

8. The line 195-196 is repetitive (same as line 184).

Response: This was a point we wanted to make sure was clear to the reader, we have revised it to not be repetitive.

9. Line 176 – snpEFF and ANNOVAR were used for annotating mutations – which information was taken?

Response: snpEFF was used to annotate synonymous and nonsynonymous mutations, indels, and frameshifts. Annovar was used to annotate exonic and non-exonic (e.g., introns, UTR) mutations, to annotate with gnomad frequencies, and cosmic information. This was added to the revised methods.

10. Line 234 – 246 – why are mutations called mutated site (only in this paragraph)?

Response: In this section we are referring to the mutated sites in each gene and TIER (ie, the specific positions of the mutations), not the counts of mutations observed at these sites.

11. Lines 363-365 rate of VAF change written as 1.32×10^{-1} , perhaps 0.13 or 13% increase would be better? What is meant by “per HPV”?

Response: We followed the reviewer’s suggestion and revised this section to use the annual VAF increase as 0.13 instead of 1.32×10^{-1} . Given that multiple HPVs can have the same hotspot mutation, we calculated the average rate of VAF change across HPVs associated with each hotspot mutation. We have revised our terminology to be more clear, we removed “per HPV” and now describe these as “Mutations with the fastest average rates of change were PIK3CA E542K ($r=0.132$ VAF increase per year) and E545K ($r=0.129$), ERBB2 R678Q ($r=0.062$), TP53 R213X ($r=0.045$), ARID1A R1446X ($r=0.034$), and PTEN Q17X ($r=0.022$).” Please see lines 386-390 in the text.

12. Line 371 and 459 – the 76 fold change is not mentioned in the results previously, although it appears in table 3. Perhaps a mention in the results, along with the other “not so striking”, but still significant fold changes would be appropriate.

Response: We agree and have added this 76-fold change to the results at lines 334-335 as suggested. Since we had a lot of results to summarize, we had only written out some of the fold change values in the results and referred the reader to the Tables/Figures for most.

13. Figure 2c TIER 2 - p value cannot be 68.

Response: Thank you for noticing that, it was a mistake, we have corrected it to $p=0.68$.

14. Figure 4, please show scatter dot plots, instead of box plots.

Response: we have revised Figure 4 to a scatter box plot, and also Supplemental Figure S4, please see new figures below. Since we are not illustrating the relationship between two continuous variables, and instead illustrating a continuous variable and a categorical variable, we think a box plot with a scatter plot is best.

New Figure 4:

New Supplemental Figure S4:

REVIEWERS' COMMENTS

Reviewer #2 (Remarks to the Author):

Thanks for thoughtfully considering this reviewer's suggestions.

Reviewer #3 (Remarks to the Author):

Dear Authors,
Thank you for answering my comments.

I have two minor points outstanding:

1. I agree with Reviewer 1 that the ratio of 1.13:1 is extremely confusing, and also does not fit any "industry standard". In your response to the reviewer, you explain how the ratio came about. The suggestion is not to exclude any controls to change the ratio to 1:1. You should definitely show all the samples that you tested, but instead of mentioning this strange ratio, you simply point the reader to a table where nCases and nControls are shown.

2. line 541: the wording "should be confirmed" should be modified to "requires validation".

**Division of Cancer
Epidemiology and Genetics**

9609 Medical Center Drive
Room 6E422
Rockville, MD 20850
Tel: (240) 276-7258

U.S. Department of
Health and Human Services
National Institutes of Health

July 9, 2024

Dr. Kathryn McGinnis,
Senior Editor, *Nature Communications*

Dear Dr. McGinnis,

Thank you for your time and effort in consideration of our manuscript entitled "Distinct somatic mutations driving carcinogenesis in 4,000 HPV-positive exfoliated cervical cell samples by HPV type" (NCOMMS-23-59346). We have addressed all editorial requests, completed the Author Checklist and Reporting Summary, and responded to the two remaining reviewer comments below.

Reviewer #3: I have two minor points outstanding:

1. I agree with Reviewer 1 that the ratio of 1.13:1 is extremely confusing, and also does not fit any "industry standard". In your response to the reviewer, you explain how the ratio came about. The suggestion is not to exclude any controls to change the ratio to 1:1. You should definitely show all the samples that you tested, but instead of mentioning this strange ratio, you simply point the reader to a table where nCases and nControls are shown.

Response: We have removed the mention of the case:control ratio and instead point the reader to the table summarizing the sample counts (Table S1).

2. line 541: the wording "should be confirmed" should be modified to "requires validation".

Response: This was revised as recommended.

Once again, thank you for your time and effort.

Sincerely,

Lisa Mirabello, PhD
Senior Investigator
Division of Cancer Epidemiology and Genetics, NCI
9609 Medical Center Drive, Room 6E422
Rockville, MD 20850
Tel: (240) 276-7258
Email: mirabellol@mail.nih.gov